# Dissociable roles of cortical excitation-inhibition balance during patch-leaving versus value-guided decisions

Luca F. Kaiser [1,2,7✉], Theo O. J. Gruendler [2,3,7], Oliver Speck[2,4,5,6], Lennart Luettgau[1,2] & Gerhard Jocham [1,2]

In a dynamic world, it is essential to decide when to leave an exploited resource. Such patch-leaving decisions involve balancing the cost of moving against the gain expected from the alternative patch. This contrasts with value-guided decisions that typically involve maximizing reward by selecting the current best option. Patterns of neuronal activity pertaining to patch-leaving decisions have been reported in dorsal anterior cingulate cortex (dACC), whereas competition via mutual inhibition in ventromedial prefrontal cortex (vmPFC) is thought to underlie value-guided choice. Here, we show that the balance between cortical excitation and inhibition (E/I balance), measured by the ratio of GABA and glutamate concentrations, plays a dissociable role for the two kinds of decisions. Patch-leaving decision behaviour relates to E/I balance in dACC. In contrast, value-guided decision-making relates to E/I balance in vmPFC. These results support mechanistic accounts of value-guided choice and provide evidence for a role of dACC E/I balance in patch-leaving decisions.

[1] Biological Psychology of Decision Making, Institute of Experimental Psychology, Heinrich Heine University, Düsseldorf, Germany. [2] Center for Behavioral Brain Sciences, Otto von Guericke University, Magdeburg, Germany. [3] Center for Military Mental Health, Military Hospital Berlin, Berlin, Germany. [4] Leibniz Institute for Neurobiology, Magdeburg, Germany. [5] German Center for Neurodegenerative Diseases (DZNE), Magdeburg, Germany. [6] Department of Biomedical Magnetic Resonance, Institute for Physics, Otto von Guericke University, Magdeburg, Germany. [7]These authors contributed equally: Luca F. Kaiser, Theo O. J. Gruendler. ✉email: kaiserl@hhu.de

In an ever-changing world with non-uniformly distributed goods, organisms have to decide whether they want to accept the resources provided by their current environment or switch to an alternative course of action. These patch-leaving decisions require balancing potential benefits in alternative environments against costs associated with abandoning the current patch (or current course of action). Patch-leaving decisions can be contrasted with value-guided choices, where agents often need to integrate multiple attributes to select the option currently most valuable. Consider for example a researcher working at a university in a small town who considers moving to Munich (famous in Germany for high cost of living). Initially, there is no benefit in leaving: moving costs money and the cost for living is higher in Munich. However, better career prospects and a higher salary might, in the long run, overcompensate the financial and social costs incurred. This constitutes a patch-leaving decision. In Munich, our researcher may face a decision on where to live—and attributes like the quality of different apartments, the rent, and the distance from the office may contribute to how valuable each flat is judged. Based on these attributes, our researcher would simply select the one they judge more valuable altogether, thus maximizing immediate reward. This kind of decision is commonly referred to as a value-based decision.

Studies in animals and humans suggest a role of the dorsal anterior cingulate cortex (dACC) in patch-leaving decisions[1–4], as well as signalling potential costs entailed by behavioural adjustments[2,3]. Activity in the dACC has been reported to reflect diminished rewards within the current environment[2,5,6] as well as the average value of potential alternatives[3] suggesting an important role in guiding behavioural adjustments[7]. In contrast, value-guided decision-making has been linked to the ventromedial prefrontal cortex (vmPFC)[8–16]. Activity in this region covaries with the values of the available options, positively with the value of the chosen option, and negatively with the value of the unchosen option[8–10,14]. Both theoretical and experimental results strongly suggest that a mechanism based on competition via mutual inhibition in vmPFC supports value-guided choice[9,15,17]. This competition is driven by the balance between GABAergic inhibition and recurrent glutamatergic excitation. Concentrations of the major excitatory and inhibitory neurotransmitters, glutamate and GABA, have been shown to be related to both choice performance and a vmPFC value comparison signal in a manner that is consistent with biophysical models[9,17]. Animal studies[2,4] suggest a similar role for the balance between glutamate and GABA in patch-leaving decisions but in dACC rather than in vmPFC[18].

We hypothesized that patch-leaving behaviour is guided by the balance between cortical excitation and inhibition (E/I balance) in dACC. In contrast, we expected that value-guided decision-making is governed by E/I balance in vmPFC. Healthy human participants performed a decision-making task (Fig. 1a) combining patch-leaving and value-based decision-making. We measured GABA and glutamate concentrations using magnetic resonance spectroscopy (MRS) at 7 T in five cortical areas of interest: vmPFC, dACC, dorsolateral prefrontal cortex (dlPFC), and bilateral primary motor cortex (M1). Specifically, we predicted that interindividual differences in how costs and patch values influence behaviour relates to variations in E/I balance in dACC[2,4,18] over and above the effects of all other voxels of interest. Further, we predicted that decision performance during value-guided choice would depend on vmPFC E/I balance[9,19]. Additionally, models based on competition by mutual inhibition predict that the speed at which a decision unfolds is driven by the available evidence, and the rate of this evidence accumulation is again crucially dependent upon E/I balance[9]. We therefore further predicted that the effect of the key decision variables on response times would also be related to E/I balance in dACC and vmPFC, respectively[20].

We report contributions of E/I balance that are dissociable as a function of decision type and cortical area. Patch-leaving behaviour is related to E/I balance in dACC but not in any of the other regions investigated. In contrast, value-guided decision-making is related to E/I balance in vmPFC but not in any of the other cortical areas.

## Results

Participants ($N = 29$) performed 320 trials of a behavioural task combining patch-leaving and value-guided choice. Each trial of the behavioural task consisted of a patch-leaving decision followed by a value-guided choice (Fig. 1a). Importantly, the task was designed such that the value-guided choice was explicitly temporally separated from the choice to leave or stay in the current patch. At the first stage, participants indicated by button press whether they wanted to stay in their current patch or leave for the alternative patch. Leaving was associated with a cost (randomly drawn from the set {5, 10, 15, 20}), which was subtracted from the participant's current total earnings. Over trials, the reward available in the current patch stochastically depleted according to a decaying Gaussian Random Walk, whereas the reward in the alternative patch replenished. The cost level was displayed to participants and remained constant until a decision to leave the patch was made, at which time a new cost level was randomly selected. Thus participants needed to monitor, over trials, the relative advantage of leaving for the alternative patch and to compare this against the cost for leaving. No money could be won at this stage. Following the patch decision stage, participants entered the value-guided choice. Here the reward available in the patch chosen by the participant was randomly divided and allocated to two choice options. Additionally, a probability with which this reward could be obtained was randomly assigned to each of these two options. This design feature ensured that the value-guided choice was temporally decorrelated from the choice to leave or stay in the current patch. While being in a rich patch will, on average, lead to better choice options at the value-guided choice stage, the exact options to choose from are not known to participants when they make their patch choice. After choice, participants received a feedback on whether their choice had been rewarded. This was followed by the next trial. In a separate session, 24–48 h after volunteers completed the behavioural task, we obtained estimates of GABA and glutamate concentrations in five cortical areas of interest (Fig. 1b and Supplementary Fig. 3) using single-voxel MRS at 7 T (see Fig. 1c for an example spectrum from one participant). We recorded from the dACC, the vmPFC, the right dlPFC, and the bilateral primary motor cortices (M1). Note that the vmPFC voxel is located in a rather dorsal position, covering parts of pregenual ACC. This location, which is also in line with previous work[9], was chosen based on methodological considerations since obtaining MRS measurements in more ventral positions is difficult due to field inhomogeneities. However, please also note that value signals in vmPFC, while not centred on this location, often extend to cover this region across a large swath of the ventral to dorsal extent of the mesial prefrontal cortex[9]. In addition to vmPFC and dACC, we selected the dlPFC because of its importance for working memory-related processes[21,22]. Since patch leaving requires carrying a representation of patch-leaving value across trials, we reasoned that dlPFC E/I balance might play a role in patch-leaving but not value-guided choice behaviour. The motor cortex was selected as a control region, where we expected a relationship with motor, but not task-specific parameters, neither value nor patch leaving related.

We used multiple linear regression to test our hypotheses. In order to limit the number of statistical comparisons, we proceeded

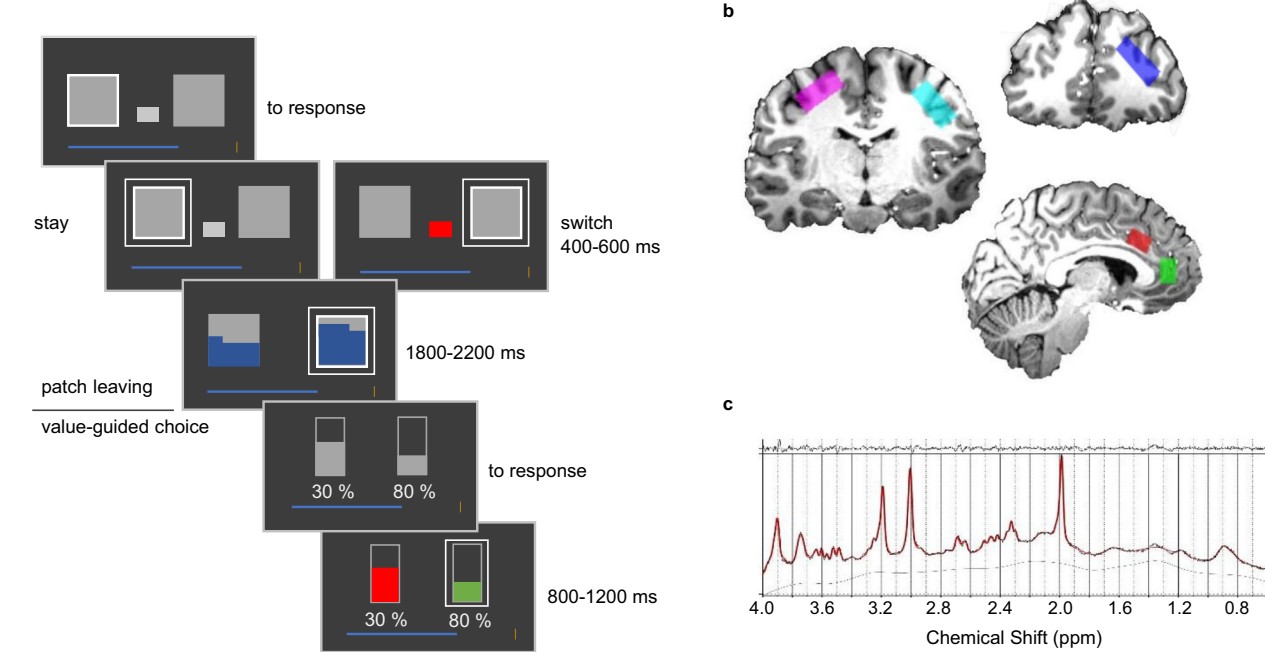

**Fig. 1 Behavioural task and MRS recordings. a** Schematic of the task structure. Participants made a patch decision in the first stage of each trial. A white outline indicated the location of the participant's current patch. If they chose to switch, they had to pay a cost indicated by the size of a grey rectangle. Following the choice, the values of both patches were revealed, indicated by the blue fillings. In the value-guided choice phase, the chosen patch value was randomly divided between the two different options and reward probabilities were randomly assigned to them. Participants selected an option by pressing a button, which was followed by feedback on both options. If they obtained a reward, the blue progress bar at the bottom of the screen grew in proportion to this reward. **b** Example MRS voxel placement for one participant (Supplementary Fig. 3 for overlay of all participants). Spectroscopy voxels were placed in right dlPFC (blue), bilateral primary motor cortex (pink and cyan), dACC (red) and vmPFC (green). **c** Spectrum obtained from dACC of one exemplary participant.

along the following hierarchy: First, we only tested behavioural variables, which we hypothesized to relate to E/I balance (see above). Second, using a general linear model (GLM), we first projected E/I balance (ratio between glutamate and GABA) from all five regions of interest onto main behavioural parameters from the patch-leaving and value-guided choice phase. All of these analyses were performed exclusively using the design matrix containing E/I balance from all five regions. Third, if, and only if, this GLM yielded a significant effect for one brain region, we followed this up by asking whether glutamate, GABA, or both within that specific region contributed to this effect of E/I balance. To this end, we then computed partial correlations, regressing out the effects of all other factors than the one currently of interest (see 'Methods'). These partial correlations can therefore be thought of as post hoc test, further investigating the individual contributions of GABA and glutamate to a main effect of E/I balance (if present).

**Patch-leaving behaviour**. Participants took costs and patch value differences into account in guiding their patch-leaving choices. On average, participants left their current patch on $20.55 \pm 1.40$ (mean ± SEM) out of 320 trials. We found that participants stayed longer in their current patch when they had to pay higher switch costs. The average (across-participants mean of the median per cost level) patch value differences (alternative − current patch) at which participants left their current patch increased with cost level (repeated-measures analysis of variance (RM-ANOVA): $F_{3,81} = 5.941$, $p = 0.001$, $\eta^2 = 0.063$, significant positive linear trend: $t_{27} = 3.961$, $p < 0.001$, confidence interval (CI)$_{95}$ = [1.341–4.223], Cohen's U3$_1$ for one sample = 0.179; Fig. 2a). To quantify how participants balanced patch values against cost, we computed a patch-leaving advantage by subtracting, for every patch-leaving trial, the switch costs from the relative benefit of leaving

(alternative − current patch value). These patch-leaving advantages were then averaged across switch trials. The average patch-leaving advantage across subjects was $19.39 \pm 2.07$ (mean ± SEM, see Fig. 2b for an evolution of patch-leaving advantages across all trials for one example participant).

To investigate the factors governing the speed of responding, we set up a multiple linear regression model. Patch value difference, patch-leaving trials, cost levels, trial number, switch (left/right) of patch presentation (relative to the previous trial), and wins in the previous trials were entered as independent variables to predict (logarithmic) response times. Participants' responses showed a trend of being slower when switching entailed greater costs ($t_{28} = 1.844$, $p = 0.076$, CI$_{95}$ = [−0.003 to 0.056], U3$_1$ = 0.345). Furthermore, they responded slower in switch trials ($t_{28} = 3.683$, $p = 0.001$, CI$_{95}$ = [0.040 to 0.138], U3$_1$ = 0.276), when they had received reward at the value-guided choice stage of the previous trial ($t_{28} = 3.733$, $p = 0.001$, CI$_{95}$ = [0.021 to 0.071], U3$_1$ = 0.207) and when there was a change in presentation sides of patch values ($t_{28} = 3.952$, $p = 0.001$, CI$_{95}$ = [0.026 to 0.082], U3$_1$ = 0.276). Further to this, participants' responding became significantly faster over the course of the experiment ($t_{28} = -7.985$, $p < 0.001$, CI$_{95}$ = [−0.317 to −0.188], U3$_1$ = 0.897). There was no significant effect of patch value difference ($t_{28} = 0.290$, $p = 0.774$, CI$_{95}$ = [−0.034 to 0.045], U3$_1$ = 0.517) on reaction times in the patch-leaving phase. Similarly, trial-wise patch-leaving advantages had no significant effect on reaction times either (see Supplementary Notes 1).

**Cortical E/I balance and patch-leaving behaviour**. We computed a patch-leaving advantage that indicates how participants balance the relative benefit expected from leaving against the cost. Regressing E/I balance against patch-leaving advantage revealed a

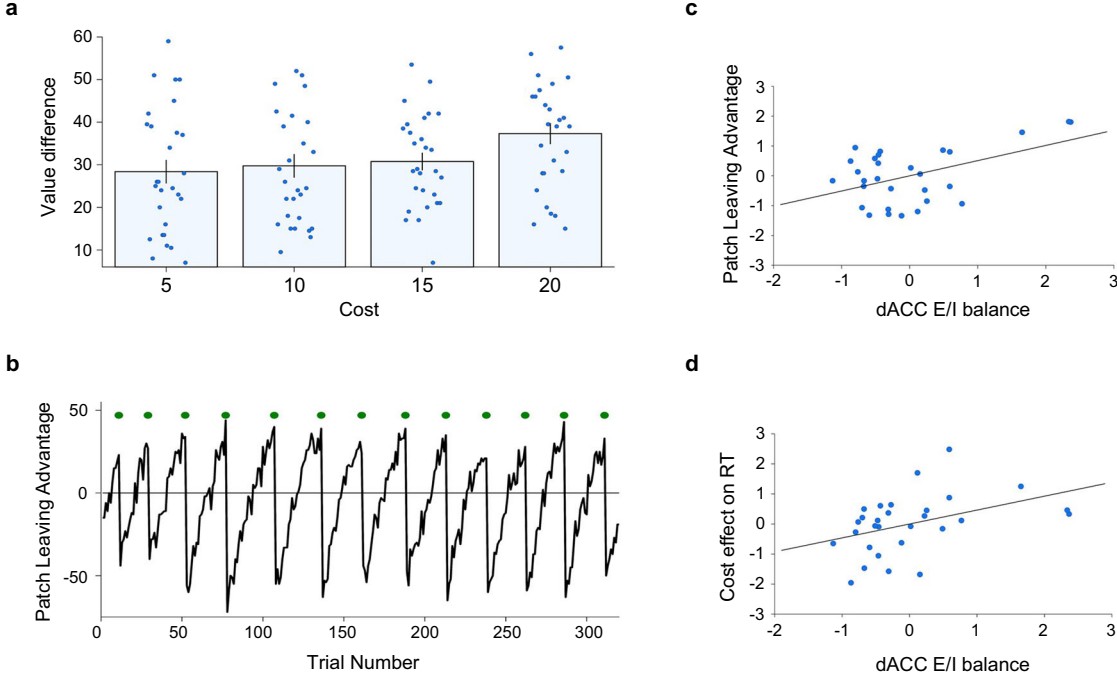

**Fig. 2 Patch-leaving behaviour and cortical E/I balance. a** Participants left their current patch at higher value differences (alternative − current patch value) when leaving was associated with higher costs (RM-ANOVA: $F_{3,81} = 5.941$, $p = 0.001$, $\eta^2 = 0.063$, $N = 28$). Individual data points are overlaid as dot plots. Bars represent across-participants mean of the median per cost level. Error bars indicate standard error of the mean. **b** Example timecourse of patch-leaving advantages (PLA) for one example participant. PLA = [value alternative patch − value current patch − cost]. Green circles indicate patch-leaving trials. **c** Participants with higher dACC E/I balance leave at higher average PLA (Pearson correlation on residuals (compare main text and methods): $r = 0.483$, $p = 0.008$, $CI_{95} = [0.141–0.722]$, $N = 29$). **d** Participants' patch-leaving decisions are slowed down with increasing cost levels, and this effect is most pronounced in participants with high levels of dACC E/I balance (Pearson correlation on residuals: $r = 0.415$, $p = 0.025$, $CI_{95} = [0.057–0.678]$, $N = 29$). Source data are provided as a Source data file.

significant influence of E/I balance in dACC ($t_{23} = 2.643$, $p = 0.015$, $CI_{95} = [0.111$ to $0.908]$; $r = 0.483$, $p = 0.008$, $CI_{95} = [0.141$ to $0.722]$, Fig. 2c) but not in any other region of interest (all $p > 0.199$). This effect was, by trend, driven by GABA in dACC ($r = -0.323$, $p = 0.087$, $CI_{95} = [-0.617$ to $0.049]$; Supplementary Fig. 1) but not glutamate in dACC ($r = -0.152$, $p = 0.431$, $CI_{95} = [-0.491$ to $0.227]$). In addition to this, a direct contrast showed that the relationship between patch-leaving advantage and E/I balance was stronger in dACC compared to both vmPFC ($t_{23} = 2.613$, $p = 0.016$) and dlPFC E/I ($t_{23} = 2.087$, $p = 0.048$). These findings suggest that the manner in which the relative benefit of leaving is balanced against travel costs is uniquely related to E/I balance in dACC but not in any of the other areas investigated.

We next assessed how cortical E/I balance was related to patch response speed and to how key task parameters affected response speed. We did not find any significant effects on overall response speed (all $p > 0.198$) but a specific effect of E/I balance in dACC on the degree to which patch decision choices were slowed by costs. Participants showed slowing of patch choices with higher cost levels, and the magnitude of this effect was related to E/I balance in dACC ($t_{23} = 2.187$, $p = 0.039$, $CI_{95} = [0.025$ to $0.900]$; $r = 0.415$, $p = 0.025$, $CI_{95} = [0.057$ to $0.678]$; Fig. 2d).

Please note that the results displayed in Fig. 2c, d primarily serve to illustrate the effects obtained in our main GLM. Nevertheless, inspection of these panels reveals three data points that appear further away from the remainder of the data. Therefore, we have additionally analysed the same data (the residuals of dACC E/I balance and behaviour) with a robust regression analysis. This confirmed the pattern of results reported above (Fig. 2c: $t_{27} = 2.665$, $p = 0.013$, $CI_{95} = [0.120$ to $0.924]$; Fig. 2d: $t_{27} = 2.034$, $p = 0.052$, $CI_{95} = [-0.004$ to $0.810]$). Thus

far, our results are consistent with our hypothesis. E/I balance in dACC, but not in any of the other regions investigated, is related both to how participants balance expected benefits against travel costs and to how costs affect the speed at which patch decisions are made.

**Value-guided choice behaviour.** Participants selected the objectively correct option (higher expected value) in 81.85 % ± 1.52 (mean ± SEM) of all trials. We set up a logistic regression model to investigate the factors that affected participants' decisions (right vs left option). Participants choices were strongly guided by the differences (right minus left) in expected values between options ($t_{28} = 6.422$, $p < 0.001$, $CI_{95} = [3.184$ to $6.166]$, $U3_1 = 0$). As expected, value sum had no significant effect on choice ($t_{28} = -1.380$, $p = 0.179$, $CI_{95} = [-0.268$ to $0.052]$, $U3_1 = 0.621$). Additionally, there was a significant effect of no-brainer trials (trials in which both probability and magnitude favoured one option) on choice ($t_{28} = 5.983$, $p < 0.001$, $CI_{95} = [0.196$ to $0.400]$, $U3_1 = 0.103$), which is likely due to an increased occurrence of no-brainer trials favouring the right option (see 'Methods'). There was no significant influence of either patch value difference, switch costs, whether the current trial was a switch trial, the current trial's patch choice (left/right), the value-guided choice from the previous trial, whether this choice had been rewarded, and of trial number (all $p > 0.486$). This indicates that participants' value-guided choices were guided by the key value-related parameters, not by other aspects, such as whether a choice had been rewarded on the previous trial.

To accumulate maximal returns, participants need to compute the Pascalian expected values by multiplying reward probabilities

and reward magnitudes and then choose the option with the higher expected value. However, humans do not weigh probabilities and magnitudes in a statistically optimal way and show systematic distortions[8,23]. We fitted several different models to explain how participants combine reward probabilities and magnitudes[23]. The models incorporated different utility functions to represent distortions in the weighting of reward information and assumed either multiplicative or additive strategies to combine reward probabilities and magnitudes. We found that choices were best explained by a model assuming multiplicative value integration with non-linear probability and magnitude weighting[23].

$$V = \omega_{mult} * (u(m_O) * w(p_O)) \qquad (1)$$

$$w(p_O) = \frac{p_O^{\gamma}}{(p_O^{\gamma} + (1 - p_O)^{\gamma})^{1/\gamma}} \qquad (2)$$

$$u(m_O) = m_O^{\alpha} \qquad (3)$$

where $m_O$ and $p_O$ are the objective reward magnitudes and probabilities that are transformed into subjective reward magnitudes and probabilities, respectively, with the shape of the functions governed by the free parameters $\alpha$ and $\gamma$. The parameter $\omega_{mult}$ scales the effect of value difference and thus corresponds to a softmax inverse temperature. For model fitting, we fixed $\omega_{mult}$ at 6.62 (see 'Methods', Supplementary Table 6, and Supplementary Information 4 for parameter recovery).

Finally, we investigated whether the same variables used to predict choices have a significant effect on normalized (log) response times. The only difference from the model used to predict binary choice (of right option) is that we used absolute expected value differences here (rather than right minus left values), since we did not expect any effect conditional on side of presentation. Participants exhibited faster responding with greater value difference between options ($t_{28} = -5.928$, $p < 0.001$, $CI_{95} = [-0.273 \text{ to } -0.133]$, $U3_1 = 0.862$). Value sum had no significant effect ($t_{28} = -1.543$, $p = 0.134$). Furthermore, participants showed faster responding on trials with high patch value difference ($t_{28} = -7.105$, $p < 0.001$, $CI_{95} = [-0.156 \text{ to } -0.086]$, $U3_1 = 1$), in no-brainer trials ($t_{28} = -14.344$, $p < 0.001$, $CI_{95} = [-0.369 \text{ to } -0.277]$, $U3_1 = 1$), and with increasing trial number ($t_{28} = -8.119$, $p < 0.001$, $CI_{95} = [-0.261 \text{ to } -0.156]$, $U3_1 = 0.931$). Finally, we found significantly slower responses in patch-leaving trials ($t_{28} = 4.590$, $p < 0.001$, $CI_{95} = [0.026 \text{ to } 0.067]$, $U3_1 = 0.276$, Fig. 3a). Neither cost levels, the previous trial's value-guided choice, nor whether reward had been received in the previous trial had an effect on reaction times in the value-guided choice phase (all $p > 0.126$). However, participants responded more slowly during value-guided choice when they had chosen the right patch during patch leaving ($t_{28} = 2.538$, $p = 0.017$, $CI_{95} = [0.005 \text{ to } 0.043]$, $U3_1 = 0.345$).

**Cortical E/I balance and value-guided choice behaviour.** To relate cortical neurochemistry to value-guided choice behaviour, we used the same approach as above for the patch-leaving phase. Similar to our previous work[9], we found that E/I balance in vmPFC was related to value-guided choice performance. Decision accuracy (percentage of choices of the higher value option) was negatively related to E/I balance in vmPFC ($t_{22} = -2.437$, $p = 0.023$, $CI_{95} = [-0.947 \text{ to } -0.076]$; $r = -0.461$, $p = 0.012$, $CI_{95} = [-0.708 \text{ to } -0.114]$; Fig. 3b) but not in any of the other regions investigated ($p > 0.406$), indicating that subjects with higher concentrations of GABA relative to glutamate were better at selecting the higher value option. When we followed this up with partial correlations, neither GABA ($r = 0.234$, $p = 0.221$) nor

glutamate alone ($r = -0.225$, $p = 0.241$) was significantly correlated with decision accuracy. These findings indicate that participants with higher concentrations of glutamate relative to GABA in vmPFC indeed tend to exhibit less accuracy in their choice behaviour. We found the same relationship with vmPFC E/I balance when we used the regression coefficients for expected value differences (that is, the degree to which participants' choices were guided by the value difference between options, see Supplementary Notes 2) instead of percentage of correct choices.

To further investigate the relationship between E/I balance and value-guided choice behaviour, we fitted a behavioural model accounting for systematic deviations in the weighting of reward information. We found that the extent to which participants distort reward magnitudes in guiding their choices (model parameter $\alpha$) was significantly related to vmPFC E/I balance ($t_{22} = -2.409$, $p = 0.025$, $CI_{95} = [-0.945 \text{ to } -0.071]$; $r = -0.457$, $p = 0.013$, $CI_{95} = [-0.705 \text{ to } -0.109]$; Fig. 3c), again without any effect of the other four regions ($p > 0.267$). This effect showed a trend of being influenced by GABA in vmPFC ($r = 0.339$, $p = 0.073$, $CI_{95} = [-0.032 \text{ to } 0.627]$). Additionally, we found a significant relationship between $\gamma$ and E/I balance in vmPFC ($t_{22} = 2.144$, $p = 0.043$, $CI_{95} = [0.015 \text{ to } 0.878]$; $r = 0.416$, $p = 0.025$, $CI_{95} = [0.058 \text{ to } 0.679]$) but not in any other region of interest (all $p > 0.327$). The effects of $\alpha$ and $\gamma$ potentially mediate the influence of vmPFC E/I balance on choice accuracy. After adding $\alpha$ and $\gamma$ to the E/I design matrix, we did not find any significant effect of vmPFC E/I balance on choice accuracy anymore ($t_{20} = 0.361$, $p = 0.722$). We found a similar effect instead when using coefficients from our logistic regression analysis, showing that a greater reliance on reward probabilities compared to magnitudes was related to E/I balance in vmPFC (Supplementary Notes 2).

Taken together, value-guided choice performance was related to E/I balance in vmPFC but not in any of the other cortical regions. Participants with high levels of GABA relative to glutamate were most reliable at selecting the higher value options, possibly due to a more optimal weighting of reward magnitudes. Finally, we asked how value-guided response speed was related to cortical E/I balance. Theoretical models predict that higher levels of inhibition will lead to more pronounced slowing on difficult decisions (trials with low value difference[9]). We first observed that overall response times in the value-guided choice phase were specifically related to dACC E/I balance ($t_{22} = -2.423$, $p = 0.024$, $CI_{95} = [-0.975 \text{ to } -0.076]$; $r = -0.459$, $p = 0.012$, $CI_{95} = [-0.707 \text{ to } -0.111]$, Supplementary Fig. 2B). This effect was contributed by a positive effect of GABA ($r = 0.452$, $p = 0.014$, $CI_{95} = [0.102 \text{ to } 0.702]$, Supplementary Fig. 2C), with no significant effect of glutamate ($r = 0.052$, $p = 0.789$). In contrast to these general effects of dACC E/I balance on overall response speed, we found a specific effect of vmPFC E/I balance on the degree to which responses were speeded up by high value difference. The effect of value difference on response times in the value-guided choice phase was lowest in individuals with high vmPFC E/I balance ($t_{22} = 2.877$, $p = 0.009$, $CI_{95} = [0.158 \text{ to } 0.972]$; $r = 0.523$, $p = 0.004$, $CI_{95} = [0.193 \text{ to } 0.746]$, Fig. 3d). Specifically, GABA levels correlated negatively with this effect ($r = -0.357$, $p = 0.057$, $CI_{95} = [-0.640 \text{ to } 0.011]$; Supplementary Fig. 2A). These findings indicate that participants' responses slowed down on difficult trials with low value difference and that this slowing was most pronounced in individuals with relatively higher levels of GABA compared to glutamate. This pattern is consistent with our previous findings showing that vmPFC decision signals emerged more rapidly with higher concentrations of glutamate and low levels of GABA[9].

In conclusion, this pattern of results mirrors the findings from the patch-leaving phase. Whereas patch-leaving behaviour was

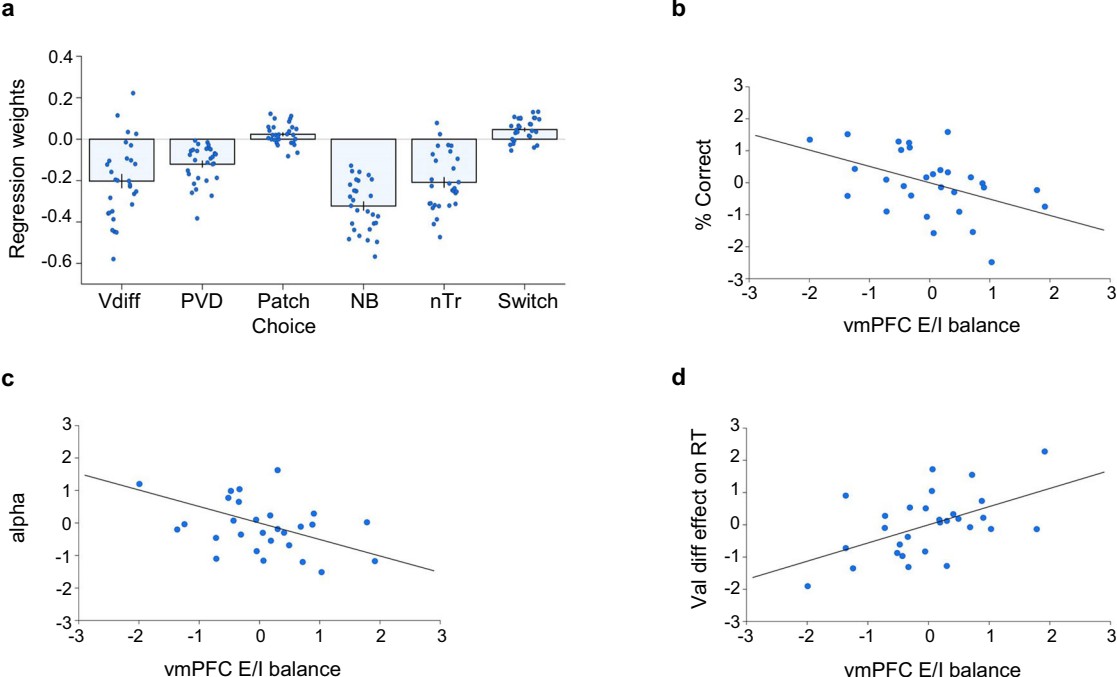

**Fig. 3 Value-guided choice behaviour and cortical E/I balance. a** Regression showing that participant's reaction times are guided by value differences between choice options (Vdiff; two-sided one-sample $t$ test against zero: $t_{28} = -5.928$, $p < 0.001$, $CI_{95} = [-0.273$ to $-0.133]$, $U3_1 = 0.862$), patch value differences (PVD; two-sided one-sample $t$ test against zero: ($t_{28} = -7.105$, $p < 0.001$, $CI_{95} = [-0.156$ to $-0.086]$, $U3_1 = 1$), choices in the patch phase (two-sided one-sample $t$ test against zero: $t_{28} = 2.538$, $p = 0.017$, $CI_{95} = [0.005$ to $0.043]$, $U3_1 = 0.345$), whether each trial is a no-brainer trial (NB; two-sided one-sample $t$ test against zero: $t_{28} = -14.344$, $p < 0.001$, $CI_{95} = [-0.369$ to $-0.277]$, $U3_1 = 1$), trial number (nTr; two-sided one-sample $t$ test against zero: $t_{28} = -8.119$, $p < 0.001$, $CI_{95} = [-0.261$ to $-0.156]$, $U3_1 = 0.931$) and whether each trial was a patch-leaving trial (Switch; two-sided one-sample $t$ test against zero: $t_{28} = 4.590$, $p < 0.001$, $CI_{95} = [0.026$ to $0.067]$, $U3_1 = 0.276$). Individual data points are overlaid as dot plots. Bars represent mean values across participants. Error bars indicate standard error of the mean. **b** Accuracy of value-guided choice is highest in participants with low levels of E/I balance in vmPFC (Pearson correlation on residuals: $r = -0.461$, $p = 0.012$, $CI_{95} = [-0.708$ to $-0.114]$). **c** Distortions in reward magnitude weighting relate to E/I balance in vmPFC (Pearson correlation on residuals: $r = -0.457$, $p = 0.013$, $CI_{95} = [-0.705$ to $-0.109]$). **d** Participants with high levels of E/I balance in vmPFC exhibit less slowing in difficult trials (Val diff = value difference) during value-guided choice (Pearson correlation on residuals: $r = 0.523$, $p = 0.004$, $CI_{95} = [0.193$ to $0.746]$). $N = 29$ in all figures. Source data are provided as a Source data file.

specifically related to dACC E/I balance, key parameters of value-guided choice behaviour were related to vmPFC E/I balance in a consistent and mechanistically plausible manner.

## Discussion

Knowing when to leave a depleting resource is a central problem for decision makers in naturalistic environments. It requires the agent to track the value of current resources, compare it to potential alternatives, and balance the potential benefits of moving against the cost incurred by moving. Within a given environment, it is crucial to consider the various attributes that jointly determine an option's value—and then to select the most valuable option in order to maximize rewards. Thus both patch-leaving and value-guided decisions are key elements of adaptive behaviour. Using a behavioural task and assessment of cortical E/I balance by MRS quantification of GABA and glutamate concentrations, we have provided evidence for a double dissociation: E/I balance in dACC, but not in any of the other regions investigated, was related to the manner in which participants balanced potential benefits of leaving against costs during patch-leaving decisions. In contrast, E/I balance in vmPFC was related to various aspects of value-guided choice.

Participants took costs into account in guiding their patch-leaving choices, as evident from the finding that they waited for higher advantages (higher value difference between current and alternative patch) as cost levels increased, and participants who

required higher advantages compared to travel costs were characterized by high dACC E/I balance. Similarly, these participants also showed stronger slowing of patch response times with increasing cost levels. An extensive literature has implicated neural activity in dACC in behavioural adjustments[24–28]. Recently, these patterns of activity have been recast in light of new evidence suggesting that dACC may encode the evidence in favour of switching away from a current default option[11]. Specifically, dACC activity contained information about the value of searching the environment for better alternatives compared to the currently available options[3]. In both primates and rodents, firing of neurons in ACC ramps up just before the animal is about to abandon its current patch and move elsewhere[2,4] or when rats abandoned current beliefs and explored alternative strategies[29]. Similarly, ACC local field potentials in the gamma range have been related to switching between exploratory and exploitative modes of behaviour[29–31]. Since gamma oscillations are driven by a balance between glutamatergic E/I by GABAergic interneurons[32–34], it was plausible for us to assume a role for cortical E/I balance in patch-leaving decisions. We found that participants with higher E/I balance (higher levels of glutamate relative to GABA) required a higher patch-leaving advantage (a higher difference between the benefits and costs expected from leaving) and showed more pronounced slowing of patch response times when costs were high. The effect of E/I balance on patch-leaving advantages was, as a trend, contributed to by GABA, but not by glutamate levels. Our findings are in line with previous reports

showing a relationship between inhibitory neurotransmission in the dACC and patch-leaving behaviour. Parvalbumin-positive GABA interneurons in rodent anterior cingulate cortex have been shown to ramp up their firing prior to the animal leaving its current patch, and firing rates of these neurons represented the animal's stay duration in the patch[4]. Furthermore, interhemispheric gamma synchronization driven by the same class of GABAergic interneurons in medial prefrontal cortex has recently been shown to enable mice to adaptively respond to changing environments[35]. Together with our results, these findings suggest that GABAergic activity in dACC may provide a signal for leaving one's current patch. Another study in primates found a similar ramping pattern in dACC neurons[2]. While the cell type from which these recordings were obtained is not known, it is likely that they were predominantly obtained from glutamatergic pyramidal cells[36,37]. This may appear contradictory at first glance but could be easily reconciled when assuming that there is an asymmetry in the proportion of neuronal pools whose activity represents the value of leaving the current patch vs those that represent the value of staying. Such an assumption is plausible given that dACC has been shown to dominantly represent value of switching away from a current strategy[3,7,38]. Under such a scenario, in a recurrently connected network with GABAergic feedback inhibition, a ramping of pyramidal cell firing would recruit feedback inhibition, which would then further increase the asymmetry between the neuronal pools, gradually favouring the pools representing the value of switching. Thus increased levels of GABAergic feedback inhibition would amplify the network transition towards favouring the alternative option and consequently bias the agent towards leaving the patch earlier. However, unlike for value-guided choice (see below), while a hypothetical model has been postulated[18], to date there exist no biophysically plausible mechanistic models for patch-leaving behaviour.

In contrast to patch-leaving decisions, value-guided choice was specifically related to E/I balance in vmPFC: high vmPFC concentrations of GABA relative to glutamate were related to an increased decision accuracy (selection of the higher value option) and a more optimal weighting of reward magnitudes. Furthermore, vmPFC GABA concentrations were also related to how participants slowed on difficult trials (choices with low value difference), with participants with high GABA concentrations again showing more pronounced slowing. These results are in line both with mechanistic models of decision-making[17,39] and our own previous findings[9]. It is thought that decisions may be generated by a mechanism that is based on competition via mutual inhibition in recurrent cortical networks that exhibit attractor dynamics[39,40]. In these models, a winner-take-all competition is implemented, where (in the binary case) activity in only one of the two pools representing the two options remains (the chosen option), whereas activity in the other pool is suppressed. One key prediction of these models is that increased GABAergic feedback inhibition slows down the attractor dynamics, allowing for more evidence to be accumulated[9,17,20]. Thus increased GABAergic tone makes decisions slower but more accurate. In our previous work, we showed that higher concentrations of GABA and low concentrations of glutamate were related to increased decision accuracy. Neurally, this was accompanied by a slower but more stable ramping of a value difference correlate in vmPFC, a neural signature of a decision[9]. Our present results match with this pattern. Choice performance was highest in participants with high vmPFC concentrations of GABA relative to glutamate, and these participants also showed the most pronounced slowing on difficult trials. Previously, we had reported a relationship between vmPFC E/I balance and choice stochasticity[9,19] whereas in the current study we find a relationship with choice accuracy. This

discrepancy is likely due to differences in task structure. Previously[9], reward magnitudes had been independent between the two options and occupied a fixed range across participants. In contrast, here, they are the result of the current patch value (with the two options' magnitudes summing up to the patch value). This has two consequences. First, trials with low magnitude difference are less likely to occur. Second, since the range of magnitudes covered is dependent on each participant's pattern of patch leaving, estimates of choice stochasticity (softmax inverse temperature) are poorly comparable across participants. Note that we also set the inverse temperature to a fixed value for fitting the models.

We found task-specific effects for patch-leaving and value-guided choice in dACC and vmPFC, respectively. However, while participants were significantly influenced by value differences between choice options during value-guided choice, we did not find any significant influence of patch value difference on reaction times during patch-leaving decisions. This discrepancy between the two stages may appear surprising at first glance. However, response times likely indicate rather different factors in the two stages. In the patch stage, participants can already make up their mind whether to switch or stay on the next trial immediately after they observe the outcome of their patch choice. In contrast, at the value-guided choice, participants cannot anticipate the options they will encounter and instead have to compute option values on the fly.

A notable aspect of our findings is that response times in each phase were modulated by events from the respective other phase. Participants' value-guided choices speeded up as the value difference between the alternative and current patch increased on trials leading up to a switch, but when participants chose to leave their patch, the immediately subsequent value-guided choice was slowed down. Conversely, patch-leaving decisions were slowed when the previous trial's value-guided choice had been rewarded. The functional significance of these effects is not clear, but the former might indicate that participants switch to a more cautious, evaluative mode of value-guided choice upon entering the alternative patch. A recent modelling account[38] suggests an interplay between dorsomedial prefrontal cortex (including dACC) and vmPFC in deciding when to switch away from ongoing behaviour, based on reliability ratings of the current strategy[41]. In our task, rewards are only obtained during value-guided choice. These potentially serve as a feedback on the current strategy, which in turn might mediate a switch from ongoing behaviour. This explanation potentially also relates to the effects of dACC E/I balance on reaction times during value-guided choice. It has been suggested previously[42] that increased glutamate levels in dACC lead participants to exploit underlying task structure, whereas increased GABA concentrations allow for learning a new model of that task.

In summary, we have shown that cortical E/I balance, as assessed by MRS quantification of baseline GABA and glutamate concentrations, is related to both patch-leaving and value-guided decision-making. We found a double dissociation, where E/I balance in dACC is related to patch-leaving, but E/I balance in vmPFC is related to value-guided choice. The pattern of results further supports models that implement a competition via mutual inhibition in recurrent cortical networks as a candidate mechanism for value-guided choice. Importantly, we provide evidence that relates dACC E/I balance to patch-leaving decisions. The pattern of results suggests that elevated GABAergic relative to glutamatergic tone in dACC may increase the propensity to switch away from a current policy. Understanding the neurochemical mechanisms underlying different types of decision-making is of potential clinical relevance, since alterations in E/I balance have been described in a number of

neuropsychiatric disorders[43,44], which are characterized by impaired decision-making behaviour[45,46].

## Methods

**Participants.** Thirty-three right-handed (Oldfield-Score[47]: 91.95 ± 1.89, mean ± SEM) male participants (age: 26.18 ± 0.65 years, mean ± SEM, range: 22–36 years) with normal ($N = 16$) or corrected to normal ($N = 17$) vision participated in this experiment. Exclusion criteria comprised a history of neurological or psychiatric illness, drug abuse, and use of psychoactive drugs or medication 24 h prior to participation. Four subjects were excluded due to excessive noise in at least one of the five spectroscopic measurements (see MR imaging (MRI) for criteria). All reported results are from the remaining $N = 29$ subjects (mean age: 26.48 ± 0.72 years, range: 22–36 years; normal vision: $N = 14$; non-smoker: $N = 22$). Written informed consent to the procedure was obtained from all subjects prior to the experiment, which was approved by the local ethics committee of the Medical Faculty of the Otto-von-Guericke-University, Magdeburg. Participants were compensated for each session and received a bonus that depended on their performance in the decision-making task.

**General study procedure.** Each participant took part in two sessions (average time between sessions: 1.52 days). The first session always involved acquisition of the decision-making task during scanning with magnetoencephalography (data not presented here); the second session involved MRS acquisition. Practical limitations prevented us from acquiring both behavioural and MRS data on the same day. However, note that MRS measures of GABA and glutamate have been reported to be stable over extended periods (weeks to months) and to be non-responsive to current task demands. Therefore, they may reflect relatively stable, trait-like properties[48–51].

**Decision-making task.** Participants were asked to maximize their rewards in a two-stage decision-making task consisting of 320 trials (Fig. 1a). They first completed 15 practice trials to familiarize themselves with the task before commencing the experiment. Stimulus presentation was controlled by Psychtoolbox 3[52,53] running on Matlab 2012b (The Mathworks Company, Natick, MA). Each trial started with presentation of the two patches (two grey squares). The patch in which the participant currently resided was indicated by a grey frame around the patch. At this stage, participants simply had to indicate by button press (with the index finger of the left or right hand, respectively) whether they wanted to stay in their current patch or switch to the alternative patch. If participants chose to leave their patch, they had to pay a travel cost indicated by the size of a grey bar presented centrally between the two patches. Travel costs were randomly drawn from the set {5, 10, 15, 20 points} and remained constant until a participant chose to leave their patch, at which stage a new cost was selected. The participant's patch choice was highlighted by a frame around the selected patch (400–600 ms, jittered). In trials where participants chose to leave, the rectangular bar signalling travel costs turned red and the costs were subtracted from a progress bar displayed below the patches that indicated the participant's total earnings. Presentation sides (left/right) of the two patches were randomly selected on each trial. Therefore, while participants could decide in advance whether they wanted to stay or leave their patch, this prevented them from preparing the actual motor response before trial onset. Afterwards, the values of the two patches (the reward available in each of them, as indicated by the blue filling) were revealed (1800–2200 ms, jittered). Importantly, the value of the participant's current patch stochastically depleted over time, whereas the alternative patch replenished. Therefore, participants were required to continually accumulate evidence in favour of abandoning their current patch. On each trial, values of the two patches were drawn from Gaussian distributions with non-stationary means and variance = 3.5. The means $\mu$ of both patches were set to 50 points initially and then diffused according to a decaying Gaussian random walk on each trial:

$$\mu_{t+1} = \lambda\mu_t + (1 - \lambda)\kappa + \varepsilon \tag{4}$$

where $\lambda$ is the decay rate that was set to 0.96, $\kappa$ is the decay centre (1 for the chosen patch, 100 for the unchosen patch), and $\varepsilon$ is zero-mean Gaussian random noise with a standard deviation = 1.2. Patch values were controlled to fall within an interval of 10–90 points. After the value of the two patches was revealed, participants entered the second stage value-guided choice. The reward available in the chosen patch was allocated to two choice options at a random ratio (ensuring that none of the two options received <10% of the total patch value and excluding a 50% split between options). Furthermore, both options were assigned a probability with which this reward could be obtained, randomly sampled from the set {0.1, 0.2,…, 0.9}. Reward probabilities were independent of each other, such that, in any given trial, either of the two options, both options, or neither of them could be rewarded. Importantly, while higher patch values will, on average, lead to better choice options, this procedure ensures that participants do not know the two choice options when they make their patch decision, thereby explicitly decoupling the patch choice from the value-guided choice. In trials where both the reward probability and magnitude of one option was higher than that of the other option (a 'no-brainer trial'), we randomly flipped the reward magnitudes of the two options in 50% of cases to control for task difficulty. Due to an error in our code,

however, this change was only applied to no-brainer trials in which the left option had higher values than the right one. Therefore, no-brainer trials were more likely to be presented on the right side of the screen and the average expected value for the right option was also higher than for the left one ($t_{18558} = -20.915$, $p < 0.001$). Reward magnitudes were indicated by the height of a grey bar and reward probabilities were presented as numbers (in percent) below each bar (Fig. 1a). Reward magnitudes were displayed relative to an outline that corresponded to the overall reward available in the current patch. Participants selected an option by pressing a button with the right or left index finger, respectively. The chosen option was highlighted by a rectangular frame and the outcome of both options was revealed (800–1200 ms, jittered). The bar representing the reward magnitude turned green if an option was rewarded in the current trial, or red otherwise. Even though participants could not benefit from knowing whether the unchosen option would have been rewarded, this procedure has proven useful to remind participants that even low probability options are occasionally rewarded and that it is beneficial to consider each option's reward probability and magnitude. Every time participants were rewarded, the progress bar grew (in proportion to the obtained magnitude) towards a goal indicated by a gold target line to the right of the screen. Outcome presentation was followed by an intertrial interval (1800–2200 ms, jittered) before participants entered the patch decision stage of the next trial. A centrally located fixation cross was present throughout the entire trial. See Supplementary Information for further details on stimulus presentation.

**Analysis of behavioural parameters.** For the patch-leaving phase, we computed, for each participant and each cost level separately, the median patch value difference (alternative − current patch) at which participants left their current patch. These average patch value differences were compared across cost levels using an RM-ANOVA. Additionally, linear trends along with a constant term were regressed against cost-level-dependent switching behaviour to estimate whether the median patch value difference increased linearly with cost level. In all cost-level-dependent analyses, the data of $N = 28$ participants were analysed since one subject was never presented with the highest cost level (cost levels were randomly assigned after each switch). Furthermore, we defined a patch-leaving advantage by subtracting travel costs from patch value differences at each patch-leaving decision. These values were then averaged across switch trials per participant. We used patch value differences from the previous trial for all analyses pertaining to the patch-leaving phase since the updated patch values are only revealed following the patch choice and hence are informative for the next trial.

For the value-guided choice phase, we computed the percentage of correct responses as the percentage of trials in which subjects chose the option with higher expected value divided by the number of trials with unequal expected value. Regression coefficients were tested against zero with a $t$ test for one sample (two sided). The MEST toolbox was used to provide estimates for effect size measures ($\eta^2$ for RM-ANOVA and Cohen's U3$_1$ for one sample to test regression coefficients against 0)[54].

**Regression analyses.** For each regression analysis, all variables were normalized ($z$-scored) and a constant term was added to each design matrix. All regressions were performed for each participant separately. Regression coefficients were tested against zero using one-sample $t$ tests (two tailed) and the MEST toolbox was used to provide estimates for effect size measures (Cohen's U3$_1$ for one sample)[54,55]. Additionally, we report the 95% CI for the mean of each distribution of regression coefficients across participants.

To analyse how key value parameters influence value-guided choice, we set up a multiple logistic regression model with choice of the left vs right (0/1) option as dependent variable. Differences and sums of expected values (the product of probabilities and magnitudes for each option) as well as patch value differences and travel costs were entered as independent variables. Here we used patch value differences from the current trial since they are already known by the participant at the time of their value-guided choice. Furthermore, we added the previous trial's value-guided choice, the current trial's patch-leaving choice, a regressor coding for whether the participant had been rewarded in the previous trial, a regressor coding whether each trial was a no-brainer trial (where both probabilities and magnitudes favour the same option), the trial number, and a regressor coding whether each trial was a switch trial or not to the design matrix. We did not run a regression model to explain choices during patch-leaving decisions, since, by design of the task, the maximal patch value difference is always reached when the participants decide to leave their current patch. The regression weights for the effects of value differences would therefore merely reflect a participant's consistency in their patch-leaving behaviour.

To analyse how various task parameters influenced response speed both in the patch-leaving and value-guided choice stage, we used multiple linear regressions with normalized log response time as the dependent variable. As above for the logistic regression, all variables were normalized, a constant was added to the design matrix, and regressions were run for each participant separately. For the patch stage, the design matrix included patch value differences from the previous trial, a binary regressor indicating whether the current trial was a switch trial (one in which the participant left their current patch), the travel cost, a regressor coding for the linear effect of trial number, and two binary regressors indicating whether the side (left/right) of patch presentation had changed with respect to the previous

trial and whether the previous trial's value-guided choice had been rewarded. We used the patch value difference from the previous trial, because the outcome of the (updated) patch values are only revealed after participants' patch choice. For the value-guided choice stage, the design matrix was the same as for the multiple logistic regression model. The only differences between models was that we used a linear link function and absolute value differences between choice options. For the patch-leaving phase (Fig. 2c, d), we have additionally analysed the residuals of dACC E/I balance and behaviour with a robust regression analyses with a bisquare weight function (tuning constant = 4.685).

**Behavioural modelling of value-guided decisions.** To formally characterize choice behaviour, we fitted several models that combined reward probabilities and magnitudes multiplicatively, additively, or as a combination of both, similar to a recently published approach[23]. All reward magnitudes were rescaled between 1 and 10 before fitting. Since it is known that humans do not weigh magnitudes and probabilities in a statistically optimal way, we considered systematic distortions in the weighting of reward information in our models ($u(m)$ and $w(p)$, for reward magnitudes and probabilities, Eqs. (5) and (6), respectively)[56].

$$w(p_O) = \frac{p_O^{\gamma}}{(p_O^{\gamma} + (1 - p_O)^{\gamma})^{1/\gamma}} \tag{5}$$

where $p_O$ are the objective reward probabilities and $\gamma$ is a free parameter used to fit subjective reward probabilities. Subjective magnitudes were estimated by:

$$u(m_O) = m_O^{\alpha} \tag{6}$$

where $m_O$ is the objective reward magnitude and $\alpha$ is a free parameter used to fit the subjective magnitude. We tested not only models with distorted values but also models where objective reward information is used. In models with objective reward information $\alpha$ and $\gamma = 1$. In all additive models, values were computed according to:

$$V = \omega_m * u(m_O) + \omega_p * w(p_O) \tag{7}$$

where $\omega_m$ is a weighting factor for reward magnitudes, and $\omega_p$ for reward probabilities. In the multiplicative models, values were computed according to:

$$V = \omega_{mult} * (u(m_O) * w(p_O)) \tag{8}$$

We also estimated a model where we fixed $\omega_{mult}$ to the median parameter across all previously recovered parameters, since we found no sufficient recovery for $\omega_{mult}$ as well as a better model fit (Supplementary Table 6) with a fixed $\omega_{mult}$ parameter.

Finally, we also estimated hybrid models as proposed earlier[23]. Here value is computed according to:

$$V = \omega_{sum} * \left( \left( 1 - \left( \frac{\omega_{mult}}{\omega_p + \omega_m + \omega_{mult}} \right) \right) * \left( \left( \frac{\omega_m}{\omega_{p+m}} \right) * u(m_O) \right. \right.$$
$$\left. \left. + \left( \frac{\omega_p}{\omega_{p+m}} \right) * w(p_O) \right) + \left( \frac{\omega_{mult}}{\omega_p + \omega_m + \omega_{mult}} \right) * (u(m_O) * w(p_O)) \right) \tag{9}$$

where $\omega_{sum} = \omega_m + \omega_p + \omega_{mult}$. We fitted each of the three different model families (hybrid, additive, and multiplicative) with distorted values for probabilities and magnitudes (SU models), with distorted values only for magnitudes but objective reward probabilities (EU models), and with objective reward magnitudes but subjective reward probabilities (EVPW models) and objective reward probabilities and magnitudes (EV models)[23]. Choice probabilites were modelled with a softmax rule based on option values. Parameters were optimized using custom-written scripts in MATLAB R2019a (The Mathworks Company, Natick, MA) and constrained non-linear optimization using MATLAB's function fmincon was used to minimize the negative log likelihood of the data given the parameters. In order to decrease the probability of fitting local minima, we used 1000 random starting points and report the combination of parameters with the lowest negative log likelihood. The Bayesian information criterion was used to compare between models. For the winning model, we simulated choices for a random set of 500 parameters for each participant and recovered parameters from these artificial data[57]. Correlations between true and recovered parameters across participants can be found in Supplementary Notes 4.

**MRS data acquisition.** MR data were acquired on a 7 T system (Siemens Healthineers) equipped with a 32-channel array head coil (Nova Medical). First, a high-resolution T1-weighted scan was acquired using an MPRAGE sequence (echo time (TE) = 2.73 ms, repetition time (TR) = 2300 ms, inversion time = 1050 ms, flip angle = 5°, bandwidth = 150 Hz/pixel, acquisition matrix = 320 × 320 × 224, voxel size = 0.8 mm³ isotropic) aligned with the anterior–posterior commissure (AC–PC). This scan was used not only for the placement of MRS voxels but also for tissue segmentation. We positioned voxels in five regions of interest, including right dlPFC, bilateral primary motor cortices (rM1 and lM1), perigenual anterior cingulate cortex within vmPFC (vmPFC/pgACC), and dACC. The dlPFC voxel was placed on the right hemisphere within the middle frontal gyrus by using the superior frontal sulcus and the inferior frontal sulcus as anatomical landmarks. We positioned the voxel as far dorsally as possible when excluding the calvaria and all

extracalvarial structures. The average dlPFC voxel centroid across participants was estimated at MNI $x = 29.79 \pm 0.85$, $y = 37.72 \pm 1.38$, $z = 24.21 \pm 1.51$ (mean ± SEM). Primary motor cortex voxels were placed on the hand knob structures, identified by their omega-like shape on the central sulcus in axial slices. Average M1 voxel centroids in standard space were estimated at MNI $x = -28.97 \pm 0.82$, $y = -18.48 \pm 0.92$, $z = 51.86 \pm 0.59$ and MNI $x = -14.76 \pm 1.06$, $z = 49.76 \pm 0.88$ for lM1 and rM1, respectively. The vmPFC voxel was mediolaterally centred on the midline and dorsoventrally on the genu of the corpus callosum, with its posterior boundary just rostral to the genu. The average voxel centroid position across subjects was estimated at MNI $x = -0.17 \pm 0.15$, $y = 41.41 \pm 1.29$, $z = 7.00 \pm 0.44$. The dACC voxel was placed with reference to the corpus callosum, the cingulate as well as surrounding sulci. We used the posterior border of the genu of the corpus callosum perpendicular to AC–PC orientation to centre the voxel (Fig. 1b)[58]. The average centroid voxel position across subjects was MNI $x = -0.07 \pm 0.19$, $y = 24.14 \pm 0.43$, $z = 29.69 \pm 0.47$ (Fig. 1b). For the MRS measurements, region-specific shimming was performed. Voxel sizes were $10 \times 20 \times 15$ mm³ for the vmPFC voxel and $10 \times 25 \times 15$ mm³ for all other voxels of interest. Afterwards, MR spectra were acquired using a stimulated echo acquisition mode (STEAM VERSE) sequence (128 averages, TR = 3000 ms, TE = 20 ms, mixing time = 10 ms, data size = 2048, bandwidth = 2800 Hz) from each voxel of interest[58].

**MR data analysis.** Spectral data were analysed using the LCModel[59]. Only metabolite measurements with a Cramér–Rao lower bound <20%, full-width half-maximum <25 Hz, and signal-to-noise ratio >8 were included. We analysed the quality of each voxel measurement using LCModel immediately after acquisition of the voxel. If one voxel did not meet the quality criteria, we repeated the acquisition of this specific voxel. We had to repeat measurement of 1 of the 5 voxels in 13 of our 29 subjects to obtain valid measurements for all 5 voxels of interest. SPM 12 (Wellcome Trust Centre for Neuroimaging, London, UK) was used to segment participants' T1-weighted anatomical images into grey matter (GM), white matter (WM), cerebrospinal fluid, soft tissue, and air/background. Each voxel's GABA and glutamate concentrations were corrected for relative GM concentrations[60] by dividing their absolute concentrations by relative GM, based on the assumption that GABA and glutamate are predominantly present in GM. As SPM 12 provides tissue probability maps, we summed across probabilities for GM for each voxel in the mask and divided by the total number of voxels within each mask to approximate relative GM. Total creatine concentrations (creatine + phosphocreatine) were normalized by the relative amount of GM and WM within each voxel ((GM+WM)/number of voxels$_{mask}$) based on the assumption that creatine is predominantly present in GM and WM. All GABA and glutamate concentrations we report are normalized by total creatine concentrations. We defined E/I balance as the ratio of (normalized) glutamate to GABA levels. Voxel masks were then interpolated to individual MRI volumes with FieldTrip[61]. To estimate average voxel centroid positions, we normalized individual volume data. More specifically, data were registered to MNI space by using tissue probability maps (TPM.nii template from SPM 12). Estimated average centroid positions were extracted from each mask in MNI space.

**Behavioural parameters and E/I balance.** For each behavioural analysis, we obtained a measure describing the individual influence of key value parameters on behaviour (decision variable). To relate decision variables to cortical E/I balance, we used multiple linear regression. In order to limit the number of comparisons, we used the following hierarchy of testing: First, we only tested those decision variables for their relationship with cortical neurochemistry (a) for which we had an a priori hypothesis and (b) that had a significant effect on behaviour. Second, using a GLM, we first projected E/I balance (ratio between glutamate and GABA) from all five regions of interest against main behavioural parameters from the patch-leaving and value-guided choice phase. All of these analyses were performed exclusively using the design matrix containing E/I balance from all five regions. Third, if, and only if, this GLM yielded a significant effect for one brain region, we followed this up by asking whether this effect of E/I balance was contributed to by glutamate, GABA, or both within that specific region. To this end, we then computed partial correlations, regressing out the effects of all other factors than the one currently of interest (see below). Fourth, when there was an effect of E/I balance on a decision variable of interest, we followed this up by analysing in more detail what aspects of a decision variable were related to E/I balance. Because of our clearly defined a priori hypotheses regarding the role of dACC vs vmPFC, we did not apply correction for multiple comparison based on the number of brain regions tested in our regression models.

We analysed the following decision variables: for the patch-leaving phase, we considered the patch-leaving advantage as the key measure of interest that describes how participants balance the expected advantage of leaving against the travel cost. For the value-guided choice phase, overall decision performance, measured as percentage of correct choices (choices of the option with higher expected value) was the primary measure of interest. The latter relationship was investigated in further detail by testing to what extent the relationship between E/I balance and % correct was driven by the individual-specific distortions of reward information as captured by the model-derived parameters $\alpha$ and $\gamma$. Finally, we assessed how E/I balance was related to the effect of relevant reward information

on reaction times. Here, for the patch-leaving phase, the effect of cost on response speed was our key parameter of interest, whereas for the value-guide choice phase, we focussed on the effect of value difference on reaction times. As a control, we also included overall response times, independent of any task parameters (Supplementary Tables 4 and 5 for an overview).

All of the E/I analyses were performed exclusively using the design matrix containing balances from all five regions. The ratios of glutamate to GABA in all voxels of interest were entered as regressors in a design matrix (along with a constant term) to predict the contributions of E/I balance onto each decision variable (one linear model per decision variable, all variables normalized). Only if a significant influence of E/I balance in a specific region was identified, we computed partial correlations. These partial correlations were first computed for E/I balance in the target region (the one showing a significant effect in the main GLM) by orthogonalizing (removing the effect of all other E/I balances) both from the decision variable and from E/I balance in the target region. These partial correlations (Pearson correlation between the residuals of decision variable and E/I balance) are what is shown in Figs. 2c, d and 3b–d. To further investigate whether a main effect of E/I balance in one region was driven by GABA or glutamate (or both), we computed further partial correlations on the orthogonal contribution of GABA and glutamate. To do so, we orthogonalized both the decision variable and the neurotransmitter of interest (GABA or glutamate, respectively) in the target area with respect to both the respective other neurotransmitter in the same region and both GABA and glutamate in all other voxels. As an example, if the GLM detected a main effect of E/I in dACC on patch-leaving advantage, and we wanted to compute the orthogonal contribution of dACC GABA to this effect, then we removed the effect of dACC glutamate and the effects of both GABA and glutamate in the other voxels from both the patch-leaving advantage and from dACC GABA. Again, we computed Pearson correlations between the residuals of the decision variable and the residuals of GABA or glutamate, respectively. In the analysis of value-guided decisions, we also controlled for the amount of no-brainer trials (mean ± SEM: 38.52% ± 0.01) as a predictor variable of no interest. We report $t$ and $p$ values for each significant regression coefficient ($p < 0.05$), testing for differences from zero, as well as $r$ and $p$ values for the subsequent partial correlations (if applicable), both with their respective 95% CIs.

**Reporting summary**. Further information on research design is available in the Nature Research Reporting Summary linked to this article.

## Data availability
The raw MRS data that support the findings of this study are available from the corresponding author upon reasonable request. Raw MRS data are not publicly available due to them containing information that could compromise research participant privacy/consent. The behavioural data and tables summarizing all MRS results are available under www.github.com/luckyluc25/ei_exp. A reporting summary for this article is available as a Supplementary Information file. Source data are provided with this paper.

## Code availability
Custom written code used to analyse the behavioural data of the current study is available under www.github.com/luckyluc25/ei_exp.

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

## Acknowledgements
This work was supported by a grant from the Deutsche Forschungsgemeinschaft (JO-787/6-1) to G.J. and by the federal state of Saxony-Anhalt and the European Regional Development Fund (ERDF 2014-2020), Vorhaben: Center for Behavioral Brain Sciences (CBBS), FKZ: ZS/2016/04/78113. The authors thank Renate Blobel-Lüer for her help with MRS recordings and Alex Waite for technical support. Computational infra-structure and support were provided by the Centre for Information and Media Tech-nology at Heinrich Heine University Düsseldorf.

## Author contributions
T.O.J.G. and G.J. designed the research. T.O.J.G. and O.S. recorded the data. L.F.K., T.O. J.G. and G.J. analysed the data. L.F.K. and G.J. wrote the manuscript. All authors dis-cussed the results at all stages of the experiment and read and edited versions and approved the final version of the manuscript.

## Funding

## Competing interests
The authors declare no competing interests.
