## [Peer Review File · Nature Communications]

Reviewers' comments:

Reviewer #1 (Remarks to the Author):

In the present paper, Kaiser and colleagues use spectroscopy to relate excitation-inhibition balance in distinct brain areas to distinct behaviours. With this technique they find a very striking dissociation between dACC (E-I related to patch leaving) and vmPFC (E-I related to value-guided choices). This work is not only novel, but also very strong methodologically: They use 7T fMRI, allowing glutamate and GABA to be measured from anatomically well circumscribed areas – overcoming problems of previous studies. Furthermore, they measure several different brain areas, allowing for dissociation of functions. The clear task design allows them to dissociate the different processes with computational model. While I have some major comments, I am quite confident that the authors will be able to do the additional analyses proposed and that they will further strengthen their findings.

Major comments:

1) Simulations.

To validate the behavioural measures, inclusion of a simulation approach would be useful; e.g. see Palminteri et al. (2017). It seems that to use behavioural measures to assess individual differences it would be worthwhile to first establish how reliably the current task allows measuring these. The plots that would be worthwhile showing are: correlations between 'ground truth' and 'fitted' parameters, as well as correlations between the fitted parameters to address whether confounding occurs between the parameters.

2) Multiple comparison correction.

On my first read of the paper, I was not sure whether the authors would need to do more corrections for multiple comparisons. However, on re-reading, I think that instead it is just an issue of how the analyses are presented that can hopefully be addressed relatively easily. Correction for multiple comparisons could be done across two dimensions: number of brain areas and number of model parameters tested. I agree with the authors that it is not necessary to correct across the number of brain areas as they had clear (and well supported from the literature) hypotheses about which brain area should relate to which behaviour. For the number of parameters tested, it might be useful to highlight the hierarchical structure of how the analyses were done more clearly: for each hypothesis (dACC -> patch leaving, vmPFC -> value guided choice), one main test could be reported that tests the hypothesis and then others that provide more refinement. This is basically what the authors already have: PLA in patch leaving for dACC and % correct choice of value-guided decisions for vmPFC. And these were then followed-up by more specific analyses. This hierarchical structure could be made clearer.

In more specific detail for each section: Patch leaving: In terms of the decisions, the PLA is used to relate to the E/I balance; but for the reaction time, instead a regression model is fit, and the effect of cost onto (log) RT is used. Is it possible to instead for RT also use PLA [or is it not given that value per se did not influence RT? If so, explain more clearly in the results]? Or vice versa, explain choices by a model with

cost and patch value separately. Overall, it would be nicer if the two analyses were more similar. One option could be to leave it in the text as is, but report these additional analyses (so that all are done for both decision and RT) in the supplements so that the reader can have a full picture.

Value-guided choice: There is a mix of model-free analyses, regression and decision models being used. There is a very nice general result: % correct in vmPFC that is very clear. The following analyses are not quite as clear in how they map onto each other (between RT and decision): From Jocham et al. 2012, I would have assumed that the inverse temperature would be tested and relate to neurotransmitter values. However, I realize that in the 2012 paper, the decision model was different; it would be interesting if it was possible – though it might not be, which would also be fine – to re-analyze the previous data with the new model to see to what extent the findings are the same or different. If they differ, this should be discussed. For the reaction times: I am not sure why ‘value sum’ only is included in the reaction time model? It would have been more analogous to the decision model if there was instead an RT model that only differs in the link function (i.e. normal distribution instead of logistic), but otherwise has the same parameters. So, overall, as for the patch leaving model, it would be nice if in the supplements there was a figure/table pulling the different analyses between RT and decision together that matches them up as well as possible and shows all tests done.

Minor comments:

3) Show specificity of voxel placement by showing heatmap of where the voxels were across subjects

4) First sentence results: specify number of participants

5) Please discuss why there was a delay of 24-48h between spectroscopy measurement and task behaviour. What does this mean for whether a state or a trait is captured?

6) The authors use a commonly used procedure of fitting each person’s behaviour with a regression. In contrast to e.g. fitting a decision model with a softmax decision rule, it is possible that general ability/motivation to do the task is captured by generally having large or small regression weights. If I understand the authors’ approach correctly, they correct for this by normalizing by the total size of the regression weights of each participants. If this is the case, please make this clearer in the text. Additionally, please show the remaining correlations between the different estimated task parameters and if they are high, consider correcting for task parameters of no interest in the analyses relating spectroscopy measurement to task measures.

7) In the figures the axis labels could be more intuitive. E.g. in Figure 2B/C, PLA could be spelled out. In figure 3A+b, ‘effect size (a.u.)’ is not very informative. In 3D, ‘lambda’ could be spelled out and in 3D, ‘delta V’)

8) One experimental study has looked at the role of glutamate receptors in value-guided decision making in humans: Scholl et al. (2015). In particular, this paper has looked at the effects of a

glutamatergic drug on parameters to do with value integration (Pascalian vs. sum). It might be worthwhile relating the findings to this.

9) In supplements include figures for each statistical test relating E/I to behaviour showing the regression coefficients for all brain areas.

10) I am a bit unsure about one sentence in the discussion: “(2) a stronger reliance on integrated value as compared to attribute comparison” This sentence sounds like it says that omega depended on E/I balance, rather than lambda?

Reviewer #2 (Remarks to the Author):

This manuscript uses an innovative behavioral paradigm to quantify and contrast different value related effects on patch foraging behavior versus value guided decision making. Quantification of these effects is then related to magnetic resonance spectroscopy (MRS) measurements of glutamate, GABA, and their ratio. The ratio of these transmitters is taken as a measure of excitation/inhibition balance. The authors report that the E/I ratio in the dACC correlates with two measures, the mean value difference between the depleted and alternative patch for each participant accounting for travel costs (what the authors refer to as the “patch leaving advantage”) and the regression coefficient summarizing the effects of patch leaving costs on reaction times. In addition, they find that the E/I ratio in vmPFC negatively predicts performance in the value-guided decision making phase of the task. Participants with a high E/I ratio are less accurate, rely more heavily on reward probabilities than magnitudes in choosing an option, and their RTs are less affected by value differences between the choice options.

While these results are potentially interesting and the behavioral modeling is well-executed, the authors’ attempt to relate the MRS measurements and behavior are methodologically flawed and therefore the manuscript is not appropriate for publication in Nature Communications. I am mainly concerned with how the authors pool the MRS measurements across the 5 voxels to predict, across individual tests, the different regression coefficients for each decision variable. They state that this was done to detect a significant effect across all regions, whereupon they feel justified to partial out the residuals for each region and then run separate correlations for each individual region. (As an aside, the authors should note/clarify that the correlation scatterplots they present are the residuals or the actual E/I ration for the dACC and vmPFC). This is an odd statistical approach because a) the authors do not indicate whether the voxel identity (i.e. region) was included as a regressor in this pooled model and b) by pooling the voxel across the subjects (5 voxels x 28 subjects) this approach artificially inflates the degrees of the freedom. Moreover, by running this analysis separately for each individual variable the authors need to correct for multiple comparisons. While it is difficult to ascertain exactly how many tests they have actually performed (they total at least 5 or more decision variables by my count) many of the marginal effects on which the novelty of the manuscript. Is based would not appear to be significant and are unlikely to replicate. A much more straightforward approach would be to run a

regression analysis for each voxel (i.e. brain region) where the coefficients for each decision variable are simultaneously entered as predictors.

Reviewer #3 (Remarks to the Author):

This study investigate the effect of E/I balance measured through MRS on the decision-making behavior in humans in a novel foraging task that involves a two-step procedure of first making a decision to stay or switch in or away from the current patch and subsequently a value-based decision, in which reward magnitude and probability are explicitly given. Positive correlation between the E/I balance in dACC with the patch leaving advantage were observed, whereas the E/I balance in vmPFC correlated negatively with the accuracy of the 2nd value-based decision within one patch. Computational modeling revealed that subject used both the distinct information about magnitude and probability as well as an integrated value signal (magnitude * probability) to make their decisions. vmPFC E/I balance also correlated positively with the parameter that balances probability and magnitude information. Furthermore, the E/I balance in both regions affect the reaction time in both decisions.

Overall, the research question is clearly defined, the experimental design is well conceived and conducted with appropriate care for details. The manuscript is well written. The discussion presents a very knowledgeable and physiologically sound interpretation of the findings and integrates a lot of seminal animal data on E/I balance. However, the computational modeling left me a bit underimpressed. The model simply replicates, what has been shown in the model-free analyses and does not elucidate the cognitive processes underlying the foraging decision in the task any further. In its current version, I would recommend this paper to a more specialized journal rather than Nature Communications with its very broad readership.

My recommendation for the paper would be a more sophisticated modeling approach to overcome the “cognitive sparseness” of the current version of the paper. The present finding on the effect of E/I balance on RTs already hints at a modulation of the speed-accuracy trade-off. Therefore, I would recommend a modeling approach along the lines of drift diffusion modeling, which is able to capture this trade-off really well. I think this approach could potentially uncover the intriguing finding that the E/I balance in a specific region may modulate the boundary separation or drift rate in these models, which would make it cognitively more interesting and meaningful for a broader audience. In addition, probability, magnitude and expected value may play a modulatory role in the framework, which could provide an informative link to the model-free results.

Minor issues / questions

1. Throughout the paper I was wondering, if the E/I balance in dACC was correlated with the one in vmPFC and whether this could explain the differential effects seen for the 1st and 2nd decision. This would be useful to include in a revised version of the manuscript.

2. I was wondering why the differential findings for GABA and Glutamate in dACC on logRT did not make it into a figure.

We are very grateful to the reviewers for their careful reading and their detailed and thoughtful comments. They have inspired a number of additional analyses and several modifications to the manuscript. We believe that the changes we made in response to the reviewers' comments and suggestions represent a substantial improvement over our initial submission.

We reproduce each of the reviewers' comments below in italics before our response to each point. Changes made to the manuscript are written in blue, both here and in the manuscript.

Reviewer 1

"[] This work is not only novel, but also very strong methodologically [...] overcoming problems of previous studies. Furthermore, they measure several different brain areas, allowing for dissociation of functions. The clear task design allows them to dissociate the different processes with computational model. While I have some major comments, I am quite confident that the authors will be able to do the additional analyses proposed and that they will further strengthen their findings."

We thank the reviewer for the very positive and careful assessment of our work, but also for spotting important points that required modifications. We hope that our answers and the corresponding changes in the revised manuscript address all of the reviewer's concerns.

1) "Simulations: To validate the behavioural measures, inclusion of a simulation approach would be useful; e.g. see Palminteri et al. (2017). It seems that to use behavioural measures to assess individual differences it would be worthwhile to first establish how reliably the current task allows measuring these. The plots that would be worthwhile showing are: correlations between ground truth and fitted parameters, as well as correlations between the fitted parameters to address whether confounding occurs between the parameters."

We thank the reviewer for pointing this out. We clearly agree that it is important to demonstrate that the fitted parameters can be retrieved from simulated data (parameter recovery). We have now run such a parameter recovery (see new section in the methods for details). In brief, we simulated 500 different sets of ground truth parameters. With this set of parameters, we generated artificial data by generating choices based on parameter dependent choice probabilities. These artificial data were then subjected to the same fitting routine as the real participant data, again randomly initializing the model at 1000 different starting points. We then used, as for the real participant data, the parameters yielding the

minimum of the negative log likelihood across all 1000 starting points and evaluated the distances between these recovered parameters and the initially fitted parameters. We now show the results of this in Supplementary Figure S4A.

Furthermore, if we correctly understand, the reviewer is also asking for the correlations between the recovered parameters. We now show the correlations in Supplementary Figure S4B. We reproduce both figures here for convenience:

Supplementary Figure S4. Overview of Simulated and Recovered Model Parameters: A) Correlation between true and recovered parameters and a histogram of the difference between true and recovered parameters for our winning model (see methods for model details). B) Correlations between recovered parameters.

In addition, we have described this procedure in the Supplementary Methods (Supplementary Analysis 4), reproduced below in blue:

4) Model Validation: Simulate and Recover

To validate our model fitting routines^{1,2}, we generated and recovered data for the model with the lowest BIC (prospect model with α and γ as free parameters). We generated 500 artificial data sets by randomly selecting α and γ parameters in the range between 0 and 3. We then recovered these parameters from the artificial data with the same procedure as used for our

real participants. We used 1000 random starting points to find the combination of free parameters yielding the minimal negative log likelihood across iterations. All fittings were done for each participant separately. The distance and correlations between recovered parameters and the ground truth parameters across subjects were estimated (Supplementary Figure S4) as well as the correlations between recovered parameters.

2a) *"Multiple Comparison Correction: On my first read of the paper, I was not sure whether the authors would need to do more corrections for multiple comparisons. However, on re-reading, I think that instead it is just an issue of how the analyses are presented that can hopefully be addressed relatively easily. Correction for multiple comparisons could be done across two dimensions: number of brain areas and number of model parameters tested. I agree with the authors that it is not necessary to correct across the number of brain areas as they had clear (and well supported from the literature) hypotheses about which brain area should relate to which behaviour. For the number of parameters tested, it might be useful to highlight the hierarchical structure of how the analyses were done more clearly: for each hypothesis (dACC -> patch leaving, vmPFC -> value guided choice), one main test could be reported that tests the hypothesis and then others that provide more refinement. This is basically what the authors already have: PLA in patch leaving for dACC and % correct choice of value-guided decisions for vmPFC. And these were then followed-up by more specific analyses. This hierarchical structure could be made clearer."*

We thank the reviewer for highlighting this. As she/he correctly points out, we tested specific predictions about the role of E/I balance in dACC and vmPFC respectively, whilst taking into account possible contributions from other regions. We fully agree with the reviewer that it would improve clarity to make the underlying hierarchical structure of the analyses more explicit. We have therefore added the following two paragraphs on page 4, 6 and pages 28 - 30 of the revised manuscript, which we reproduce below in blue for convenience.

Introduction:

We hypothesized that patch-leaving behaviour is guided by the balance between cortical excitation and inhibition (E/I balance) in dACC. In contrast, we expected that value-guided decision making is governed by E/I balance in vmPFC. Healthy human participants performed a novel decision-making task (Figure 1A) combining patch-leaving and value-based decision-making. We measured GABA and glutamate concentrations using magnetic resonance spectroscopy (MRS) at 7T in five cortical areas of interest: vmPFC, dACC, dorsolateral prefrontal cortex (dlPFC), and bilateral primary motor cortex (M1). Specifically, we predicted that interindividual differences in how costs and patch values influence

behaviour relates to variations in E/I balance in dACC³⁻⁵ over and above the effects of all other voxels of interest. Further, we predicted that decision performance during value-guided choice would depend on vmPFC E/I balance^{6,7}. Additionally, models based on competition by mutual inhibition predict that the speed at which a decision unfolds is driven by the available evidence, and the rate of this evidence accumulation is again crucially dependent upon E/I balance⁶. We therefore further predicted that the effect of the key decision variables on response times would also be related to E/I balance in dACC and vmPFC, respectively.⁸

We report contributions of E/I balance that were dissociable as a function of decision type and cortical area. Patch leaving behaviour was related to E/I balance in dACC, but not in any of the other regions investigated. In contrast, value-guided decision making was related to E/I balance in vmPFC, but not in any of the other cortical areas.

Results:

We used multiple linear regression to test our hypotheses. In order to limit the number of statistical comparisons, we proceeded along the following hierarchy: Firstly, we only tested behavioural variables which we hypothesized to relate to E/I balance (see above). Secondly, using a general linear model (GLM), we first projected E/I balance (ratio between glutamate and GABA) from all five regions of interest onto main behavioural parameters from the patch-leaving and value-guided choice phase. All of these analyses were performed exclusively using the design matrix containing E/I balance from all five regions. Thirdly, if, and only if, this GLM yielded a significant effect for one brain region, we followed this up by asking whether glutamate, GABA, or both within that specific region contributed to this effect of E/I balance. To this end, we then computed partial correlations, regressing out the effects of all other factors than the one currently of interest (see methods). These partial correlations can therefore be thought of as post hoc test, further investigating the individual contributions of GABA and glutamate to a main effect of E/I balance (if present).

Methods:

For each behavioural analysis, we obtained a measure describing the individual influence of key value parameters on behaviour ("decision variable"). To relate decision variables to cortical E/I balance, we used multiple linear regression. In order to limit the number of comparisons, we used the following hierarchy of testing: Firstly, we only tested those decision variables for their relationship with cortical neurochemistry (a) for which we had an a priori hypothesis and (b) that had a significant effect on behaviour. Secondly, using a general linear model (GLM), we first projected E/I balance (ratio between glutamate and GABA) from all five regions of interest against main behavioural parameters from the patch-leaving and value-guided choice phase. All of these analyses were performed exclusively using the

design matrix containing E/I balance from all five regions. Thirdly, if, and only if, this GLM yielded a significant effect for one brain region, we followed this up by asking whether this effect of E/I balance was contributed to by glutamate, GABA, or both within that specific region. To this end, we then computed partial correlations, regressing out the effects of all other factors than the one currently of interest (see below). Fourthly, when there was an effect of E/I balance on a decision variable of interest, we followed this up by analysing in more detail what aspects of a decision variable were related to E/I balance.

We analysed the following decision variables: for the patch leaving phase, we considered the patch leaving advantage as the key measure of interest that describes how participants balance the expected advantage of leaving against the travel cost. For the value-guided choice phase, overall decision performance, measured as % correct choices (choices of the option with higher expected value) was the primary measure of interest. The latter relationship was investigated in further detail by testing to what extent the relationship between E/I balance and % correct was driven by the individual-specific underweighting of reward magnitudes as captured by the model-derived parameters α and γ . Finally, we assessed how E/I balance was related to the effect of relevant reward information on reaction times. Here, for the patch-leaving phase, the effect of cost on response speed was our key parameter of interest, whereas for the value-guide choice phase, we focussed on the effect of value difference on reaction times. As a control, we also included overall response times, independent of any task parameters (Supplementary Table 4 and 5 for an overview).

All of the E/I analyses were performed exclusively using the design matrix containing balances from all five regions. The ratios of glutamate to GABA in all voxels of interest were entered as regressors in a design matrix (along with a constant term) to predict the contributions of E/I balance onto each decision variable (one linear model per decision variable, all variables normalized). In addition to t- and p- values we report the 95 % confidence intervals for each significant ($p < 0.05$) regression coefficient. Only if a significant influence of E/I balance in a specific region was identified, we computed partial correlations. These partial correlations were first computed for E/I balance in the target region (the one showing a significant effect in the main GLM) by orthogonalizing (removing the effect of all other E/I balances) both from the decision variable and from E/I balance in the target region. These partial correlations (Pearson correlation between the residuals of decision variable and E/I balance) are what is shown in 2 C, D and 3 B, C, D. To further investigate whether a main effect of E/I balance in one region was driven by GABA or glutamate (or both), we computed further partial correlations on the orthogonal contribution of GABA and glutamate. To do so, we orthogonalized both the decision variable and the neurotransmitter of interest

(GABA or glutamate, respectively) in the target area with respect to both the respective other neurotransmitter in the same region and both GABA and glutamate in all other voxels. As an example, if the GLM detected a main effect of E/I in dACC on patch-leaving advantage, and we wanted to compute the orthogonal contribution of dACC GABA to this effect, then we removed the effect of dACC glutamate and the effects of both GABA and glutamate in the other four voxels from both the patch-leaving advantage and from dACC GABA. Again, we computed Pearson correlations between the residuals of the decision variable and the residuals of GABA or glutamate, respectively. In the analysis of value-guided decisions, we also controlled for the amount of no brainer trials (mean \pm SEM: 38.52 % \pm 0.01) as a predictor variable of no interest. We report the correlation results along with their respective p-values and 95 % confidence intervals for each correlation coefficient. We report the respective r- and p- values for each correlation as well as the 95 % confidence interval for each regression coefficient.

2b) "In more specific detail for each section: Patch leaving: In terms of the decisions, the PLA is used to relate to the E/I balance; but for the reaction time, instead a regression model is fit, and the effect of cost onto (log) RT is used. Is it possible to instead for RT also use PLA [or is it not given that value per se did not influence RT? If so, explain more clearly in the results]?"

Thank you for your remark as we had so far not sufficiently elaborated on that. In the regression, we did include both patch value difference and cost as factors that might affect RT. However, while cost did have a statistically significant effect on RT, patch value difference did not. Therefore, while this analysis would be more consistent, any relationship between cost and RT would be masked by the non-effect of patch value difference.

In addition, following the reviewer's suggestion, we now present the same analysis with PLA in the Supplementary Analysis 1 (reproduced below in blue). The reason that cost did influence RT while patch values did not is something we hypothesize to be related to the structure of the task. While patch values have to be remembered from the outcome of the last trial's patch choice, the current leaving cost is explicitly presented at the outset of each trial.

1) Reaction times in the patch leaving phase are not influenced by trial-wise patch leaving advantages

We reran a regression model to analyze reaction times during patch-leaving decisions. Here, we included trial-wise patch leaving advantages instead of using costs and patch value differences as separate regressors. This analysis revealed no significant influence of PLA ($t_{28} = -0.118$, $p = 0.907$, $CI_{95} = [-0.042 - 0.038]$, $U_3 = 0.483$). We again find a significant influence

of whether each trial was a switch trial or not ($t_{28} = 3.776$, $p = 0.001$, $CI_{95} = [0.042 - 0.141]$, $U_3 = 0.310$), of trial number ($t_{28} = -8.039$, $p < 0.001$, $CI_{95} = [-0.310 - -0.184]$, $U_3 = 0.897$), of whether the presentation side of patch values changed with respect to the last trial ($t_{28} = 3.844$, $p = 0.001$, $CI_{95} = [0.025 - 0.083]$, $U_3 = 0.276$) and of whether the value-guided choice in the last trial had been rewarded ($t_{28} = 3.689$, $p = 0.001$, $CI_{95} = [0.020 - 0.071]$, $U_3 = 0.207$). The finding that costs did influence reaction times whereas neither PLA nor patch value differences (analysis in main text) had an effect is likely related to the structure of the task. The cost of leaving is displayed on screen at the outset of each trial, whereas patch values have to be held in memory from the outcome of the last trial's patch choice.

2c) "Or vice versa, explain choices by a model with cost and patch value separately. Overall, it would be nicer if the two analyses were more similar. One option could be to leave it in the text as is but report these additional analyses (so that all are done for both decision and RT) in the supplements so that the reader can have a full picture."

Thank you for pointing this out. Indeed, exactly as suggested by the reviewer, we had originally intended to analyse choices using a regression model too. However, by design of the task, the patch value difference always increases up to the point at which the participant leaves, from when on the difference declines again. In other words, participants always switch away from their patch when patch value difference is greatest. An effect of patch value difference on choice would therefore mainly reflect a participant's consistency across patch-leaving decisions. We have added the following brief paragraph to our methods section:

We did not run a regression model to explain choices during patch-leaving decisions since, by design of the task, the maximal patch value difference is always reached when the participants decide to leave their current patch. The regression weights for the effects of value differences would therefore merely reflect a participant's consistency in their patch leaving behaviour.

2d) "Value-guided choice: There is a mix of model-free analyses, regression and decision models being used. There is a very nice general result: % correct in vmPFC that is very clear. The following analyses are not quite as clear in how they map onto each other (between RT and decision): From Jocham et al. 2012, I would have assumed that the inverse temperature would be tested and relate to neurotransmitter values. However, I realize that in the 2012 paper, the decision model was different; it would be interesting if it was possible; though it might not be, which would also be fine to re-analyze the previous

data with the new model to see to what extent the findings are the same or different. If they differ, this should be discussed."

We thank the reviewer for this important point. In our 2012 paper we had related E/I balance to the softmax inverse temperature, since this is particularly a reflection of how well participants perform on difficult trials (with low value difference). We had therefore reasoned that it would be a more sensitive behavioural measure than % correct choices, for which we had indeed not observed any correlation with either GABA or glutamate. In the 2012 study, we had used a multiplicative model in which reward probabilities and magnitudes were distorted according to prospect theory. While this kind of model can be considered good practice at that time, recent reports suggest that humans (and non-human primates) may either deploy additive or multiplicative strategies, or a hybrid of both, depending on particulars of the task structure (Farashahi et al., 2019). We have now implemented the twelve different models described in Farashahi et al. (2019)⁹ and applied them both to our current data and those from the 2012 study. We have added these results as well as a discussion of these findings to the Supplementary Materials (Exploratory Findings: Relating current finding to own previous work). In brief, in the 2012 data, the winning model is a hybrid model (a weighed combination of both an additive and a multiplicative component) without distortion of either magnitude or probability. In the current data, the winning model is a purely multiplicative model that incorporates distortion of both magnitudes and probabilities with a fixed choice stochasticity parameter. There are a number of possible reasons for this. Firstly, in the 2012 data, the trials' combination of reward attributes had been specifically optimized (offline) for the value-guided choice task to allow a certain level of difficulty, to control for correlation between chosen and unchosen value and to incorporate a certain range of no brainer trials. In contrast, in the current task, reward magnitudes are generated from the chosen patch, a random fraction of which is allocated to the two patches. As a consequence, trials with low value difference are less frequent. Secondly, in the current task the distortion of reward magnitudes becomes more important since magnitudes can potentially cover a wider range of values that depends on the current patch value, as opposed to a fixed minimum and maximum in the 2012 study. Please see our new discussion in the Supplementary Materials (Exploratory Findings: Relating current finding to own previous work) for more detail on this.

A consequence of the new modelling approach used is that we can no longer relate E/I balance to softmax (inverse) temperature, since our winning model does not contain an explicit temperature parameter anymore. In our current dataset however, we now observe a relationship between vmPFC E/I and magnitude distortion alpha, such that individuals with

higher levels of GABA weight magnitudes more optimally (less underweighting of reward magnitudes). To make our results independent of the particular winning model, we also used regression to estimate the impact of reward magnitudes, probabilities, and their multiplicative combination on choice. Replicating the model-derived results, we also find that the degree to which participants' choices are guided by reward magnitudes is related to vmPFC E/I balance (see Supplementary Material 2). Overall, we find it very reassuring that our main finding from the value-guided choice phase, the relationship between vmPFC E/I and % correct choice is entirely insensitive to any particular modelling choice.

2e) "For the reaction times: I am not sure why value sum only is included in the reaction time model? It would have been more analogous to the decision model if there was instead an RT model that only differs in the link function (i.e. normal distribution instead of logistic), but otherwise has the same parameters. So, overall, as for the patch leaving model, it would be nice if in the supplements there was a figure/table pulling the different analyses between RT and decision together that matches them up as well as possible and shows all tests done."

In our previous version of the manuscript, we had not included value sum to the decision model because, in our view, there was no reason to expect that the value sum of the two options would bias choice of the right vs left option. Nevertheless, we agree with the reviewer that it would be more coherent to adopt the same model for both analyses. In the revised manuscript, we have therefore changed the analyses accordingly, explaining choices and reaction times with the same design matrix (but different link function). The design matrix now includes a constant, difference in expected values between options, sum of expected values, patch leaving costs, patch value differences, choices in the last trial's value-guided choice, a binary regressor coding whether a participant had received reward in the previous trial, motor response (left/right choice) in the patch leaving phase, a binary regressor coding for no brainer trials (both magnitude and probability are greater for one option), trial number, and a binary vector indicating whether the trial is a patch leaving trial or not. The only difference between models is that we have used absolute differences between expected values to explain reaction times but differences between right and left option to explain choices. We have changed the results accordingly:

Choice:

We set up a logistic regression model to investigate the factors that affected participants' decisions (right vs. left option). Participants choices were strongly guided by the differences (right minus left) in expected values between options ($t_{28} = 6.422$, $p < 0.001$, $CI_{95} = [3.184 -$

6.166], $U_3 = 0$). As expected, value sum had no significant effect on choice ($t_{28} = -1.380$, $p = 0.179$, $CI_{95} = [-0.268 - 0.052]$, $U_3 = 0.621$). Additionally, there was a significant effect of no brainer trials (trials in which both probability and magnitude favoured one option) on choice ($t_{28} = 5.983$, $p < 0.001$, $CI_{95} = [0.196 - 0.400]$, $U_3 = 0.103$), which is likely due to an increased occurrence of no brainer trials favouring the right option (see methods). There was no significant influence of either patch value difference, switch costs, whether the current trial was a switch trial, the current trial's patch choice (left/right), the value-guided choice from the previous trial, whether this choice had been rewarded, and of trial number (all $p > 0.486$). This indicates that participants' value-guided choices were guided by the key value-related parameters, not by other aspects, such as whether a choice had been rewarded on the previous trial.

RT:

Finally, we investigated whether the same variables used to predict choices have a significant effect on normalized (log) response times. The only difference from the model used to predict binary choice (of right option) is that we used absolute expected value differences here (rather than right minus left values), since we did not expect any effect conditional on side of presentation. Participants exhibited faster responding with greater value difference between options ($t_{28} = -5.928$, $p < 0.001$, $CI_{95} = [-0.273 - -0.133]$, $U_3 = 0.862$). Value sum had no significant effect ($t_{28} = -1.543$, $p = 0.134$). Furthermore, participants showed faster responding on trials with high patch value difference ($t_{28} = -7.105$, $p < 0.001$, $CI_{95} = [-0.156 - -0.086]$, $U_3 = 1$), in no brainer trials ($t_{28} = -14.344$, $p < 0.001$, $CI_{95} = [-0.369 - -0.277]$, $U_3 = 1$) and with increasing trial number ($t_{28} = -8.119$, $p < 0.001$, $CI_{95} = [-0.261 - -0.156]$, $U_3 = 0.931$). Finally, we found significantly slower responses in patch leaving trials ($t_{28} = 4.590$, $p < 0.001$, $CI_{95} = [0.025 - 0.067]$, $U_3 = 0.276$, Figure 3A). Neither cost levels, the previous trial's value-guided choice, nor whether reward had been received in the previous trial had an effect on reaction times in the value-guided choice phase (all $p > 0.126$). However, participants responded more slowly during value-guided choice when they had chosen the right patch during patch leaving ($t_{28} = 2.538$, $p = 0.017$, $CI_{95} = [0.005 - 0.043]$, $U_3 = 0.345$).

Minor Comments

M1) "Show specificity of voxel placement by showing heatmap of where the voxels were across subjects."

We thank the reviewer for this suggestion. The heatmaps are now added to Supplementary Figure S3 and reproduced here below for the reviewer's convenience.

Supplementary Figure S3. Overlay of voxel placements across all participants: Average locations of all regions of interest. Brighter colors indicate a greater overlap across participants. Left: Average placement of dlPFC voxel. Middle: Average location of M1 voxels. Right: Average location of vmPFC and dACC MRS voxel.

M2) "First sentence results: specify number of participants"

We have added the following to the first sentence of the results section: Participants (N = 29) performed 320 trials of a novel behavioural task combining patch leaving and value-guided choice [..]

M3) "Please discuss why there was a delay of 24-48h between spectroscopy measurement and task behaviour. What does this mean for whether a state or a trait is captured?"

Thank you for your careful reading. It is important that this information is provided. We have added the following paragraph to the discussion:

Practical limitations prevented us from acquiring both behavioural and MRS data on the same day. However, note that MRS measures of GABA and glutamate have been reported to be stable over extended periods (weeks to months) and to be non-responsive to current task demands. Therefore, they may reflect relatively stable, trait-like properties¹⁰⁻¹³.

M4) "The authors use a commonly used procedure of fitting each person behaviour with a regression. In contrast to e.g. fitting a decision model with a softmax decision rule, it is possible that general ability/motivation to do the task is captured by generally having large or small regression weights. If I understand the authors approach correctly, they correct for this by normalizing by the total size of the regression weights of each participants. If this is the case, please make this clearer in the text."

We thank the reviewer for pointing this out. In the first version of our manuscript we had indeed normalized each participant's regression weights by the norm of their regression weight vector before testing regression coefficients against zero. However, we realized that this approach increases interdependencies among coefficients since variance in regression weights is explained across and not within participants in further analyses. In the current version of the manuscript we have therefore revised this approach and normalized (z-scored) the entire design matrix, and, for the linear regression also z-scored the dependent variable. We have changed the respective paragraph in the Methods (Regression analyses).

M5) "Additionally, please show the remaining correlations between the different estimated task parameters and if they are high, consider correcting for task parameters of no interest in the analyses relating spectroscopy measurement to task measures."

We now report correlations between all variables of interest (separately for the patch and value-guided phase) in the new Supplementary Tables 1 and 2. Please note that one should not necessarily expect them to be uncorrelated in all cases, since they may tap into the same mechanism. For instance, consider (value-guided choice) the case for % correct on one hand and the effect of value difference on RT. As you can see from our Supplementary Table 2, there is a relatively high negative correlation between these two variables, indicating that the (negative) effect of value difference on RT is most pronounced in participants with high percentage of correct choices. This, however, is exactly what would be mechanistically predicted from models using competition via mutual inhibition: Slowing the decision in the face of a lot of noise (a difficult trial with low value difference) allows for the choice to be dominated by the available evidence, while averaging out (neural) noise over time. This said however, we think it is important to account for these correlations, as the reviewer suggests, to avoid spuriously reporting relationships with E/I balance. We have therefore added Supplementary Analysis 3, where we take the reverse approach as before: rather than accounting for the contribution of other brain areas, we now account for the orthogonal contributions of the individual parameters of interest by putting them into one design matrix and regressing them against E/I balance in either dACC or vmPFC.

M6) "In the figures the axis labels could be more intuitive. E.g. in Figure 2B/C, PLA could be spelled out. In figure 3A+b effect size (a.u.) is not very informative. In 3D, lambda could be spelled out and in 3D, delta V)."

We now spell out PLA and changed the effect size label to "regression coefficients". Additionally, we changed ΔV effect on RT to "Vdiff effect on RT" (lambda is not reported anymore in the revised manuscript).

M7) "One experimental study has looked at the role of glutamate receptors in value-guided decision making in humans: Scholl et al. (2015). In particular, this paper has looked at the effects of a glutamatergic drug on parameters to do with value integration (Pascalian vs. sum). It might be worthwhile relating the findings to this."

We thank the reviewer for pointing us to the important work by Scholl et al. (2015). Clearly, this study is of high relevance for our findings. When looking at our new model comparison, the hybrid model with linear weighting of magnitudes and probabilities performs worse (higher BIC) compared to the multiplicative model with subjective values (distorted magnitudes and probabilities). We have therefore not included the effects of E/I balance on optimal linear value integration in the revised version of our manuscript. However, to relate our findings to those of Scholl et al., we ran a set of exploratory analyses. When we relate E/I balance to the model parameters of a linear hybrid model, we find an increased reliance on multiplicative value integration with lower vmPFC E/I balance ($t_{22} = -2.423$, $p = 0.024$) as well as a reduced reliance on magnitude compared to probability values within the additive module ($t_{22} = -2.711$, $p = 0.013$) (compare Supplementary Material: Exploratory Findings: Relating current finding to own previous work) for further details. We also re-analyzed the data from Jocham et al. (2012)⁶. Here, we found that a linear hybrid model (one without distortion of magnitudes and probabilities) provided the best account for the data (Supplementary Table 7). However, we did not find a relationship between E/I balance in vmPFC and multiplicative versus additive value updating. We have added these exploratory analyses and a discussion of our findings to the Supplementary Analysis.

M8) "In supplements include figures for each statistical test relating E/I to behaviour showing the regression coefficients for all brain areas."

Thanks for your suggestion. We have now added table Supplementary Table 4 and 5 'Overview of neurochemical effects for all regression models' to the revised Supplementary Materials.

M9) "I am a bit unsure about one sentence in the discussion: (2) a stronger reliance on integrated value as compared to attribute comparison; This sentence sounds like it says that omega depended on E/I balance, rather than lambda?"

The reviewer is absolutely correct. In the previous version of our manuscript we had indeed reported a relationship of both omega and lambda with vmPFC E/I balance. We have updated this paragraph in accordance with our new modelling results. However, in our new set of models, a standard prospect model with a fixed inverse temperature parameter fits the data best. We have therefore not included the effects E/I on omega in the revised manuscript. Nevertheless, as reported above, we have analysed the reliance on multiplicative versus additive value integration to relate our findings to previous publications. Here, we could confirm our previous findings that participants rely less on integrated values with higher vmPFC E/I balance.

Reviewer 2

"This manuscript uses an innovative behavioral paradigm to quantify and contrast different value related effects on patch foraging behavior versus value guided decision making. Quantification of these effects is then related to magnetic resonance spectroscopy (MRS) measurements of glutamate, GABA, and their ratio. [...] While these results are potentially interesting and the behavioral modeling is well-executed, the authors attempt to relate the MRS measurements and behavior are methodologically flawed []."

We thank the reviewer for her/his assessment of our work. Upon reading their comments, we were left with the impression that the reviewer actually quite much liked our study and the results we presented, but unfortunately the reviewer is concerned about the validity of our analysis approach. However, after careful reading of the comments, we think there is simply a misunderstanding, likely because, in our previous version, we had not explained our analysis approach and the underlying logic very well. In fact, we would argue that the exact concerns the reviewer is raising are indeed very carefully and conservatively controlled for. We detail this below in response to the individual comments and hope that, after reading this and the revised paragraphs of our manuscript, the reviewer will agree that there was a misunderstanding and that our findings are valid and obtained using sound methodology.

1) *"While these results are potentially interesting and the behavioral modeling is well-executed, the attempt to relate the MRS measurements and behavior are methodologically flawed and therefore the manuscript is not appropriate for publication in Nature Communications. I am mainly concerned with how the authors pool the MRS measurements across the 5 voxels to predict, across individual tests, the different regression coefficients for each decision variable. They state that this was done to detect a significant effect across all regions, whereupon they feel justified to partial out the residuals for each region and then run separate correlations for each individual region (As an aside, the authors should note/clarify that the correlation scatterplots they present are the residuals or the actual E/I ratio for the dACC and vMPFC). This is an odd statistical approach because a) the authors do not indicate whether the voxel identity (i.e. region) was included as a regressor in this pooled model and b) by pooling the voxel across the subjects (5 voxels x 28 subjects) this approach artificially inflates the degrees of the freedom. Moreover, by running this analysis separately for each individual variable the authors need to correct for multiple comparisons. While it is difficult to ascertain exactly how many tests they have actually performed (they total at least 5 or more decision variables by my count) many of the marginal effects on which the novelty of the manuscript is based would not appear to be significant and are unlikely to replicate. A much more straightforward approach would be to run a regression analysis for each voxel (i.e. brain region) where the coefficients for each decision variable are simultaneously entered as predictors."*

If we understand the reviewer correctly, there are three concerns, the first related to "pooling MRS measurements across the 5 voxels", the second to how we computed partial correlations, and the third to the number of tests performed and the resulting multiple comparisons.

1a) "Pooling": We would like to elaborate on our approach in more detail, as this might make the reviewer's concern on "pooled measurements" or the inclusion of an extra regressor indicating voxel identity obsolete. We have also included a section in the methods explaining this, which we reproduce below in blue. For each of the five voxels, we have divided glutamate by GABA concentrations. This was our index of E/I balance. For our regression models, we then took this value of E/I balance for each voxel and each of the 29 participants. Thus, the resulting design matrix has 29 rows and five columns (plus a constant), where each row contains one particular participant's E/I balance for all the five cortical regions. The reviewer suggests adding an extra regressor coding for the voxels' identity - however, the voxel's identity already is defined by their corresponding column. As an example, if column

three of the design matrix contains E/I balance in vmPFC, and there is an effect of column 3, there is no further need to indicate the voxel's identity (indeed, such a regressor would amount to a column of ones for each of the five regions, as they are present in all individuals). We apologize for not having spelt that out clearly enough in our previous version. We hope that our above explanation, together with the new paragraph, helped to clarify this.

When the reviewer says that we "pooled MRS measurements", is it possible that they meant because we used the E/I ratio, rather than separately entering GABA and glutamate concentrations? If so, again we could not see any perspective from which this could be problematic. To the contrary: first, whether or not one decides to merge several measures into one aggregate measure is, in our view, mostly a question of whether this is conceptually sensible. To compute a measure that is physiologically plausible measure like E/I balance certainly is - our whole reasoning and research question is based on the balance between excitation and inhibition^{6,14,15}. If we had instead divided neurotransmitters by, say lactate concentrations, we might obliterate variance and miss effects of interest. Similarly, the statement that this procedure would "artificially inflate degrees of freedom" is not evident to us, unless the reviewer means that having fewer columns in the design matrix makes estimation of the coefficients more robust, with which we clearly agree. Since computation of the coefficients' variance estimates involves multiplication by $(n-k)$, where n = number of rows and k = number of columns, estimation of the coefficients is more precise. Hence, we would argue that the regression is statistically more powered than a model with more columns and the same number of data points (rows). Therefore, in this case we would respectfully disagree with the reviewer that our approach might artificially increase the occurrence of significant results.

1b) Partial correlations: *"They state that this was done to detect a significant effect across all regions, whereupon they feel justified to partial out the residuals for each region and then run separate correlations for each individual region"*.

Our apologies, we believe this again was not clearly explained and we have now re-written the corresponding paragraphs (reproduced below in blue). Importantly, we had *not* run separate correlations for each individual region. If, and only if, our main GLM yielded an effect of E/I balance in region X, then we further investigated two partial correlations: The partial (see below) correlations of GABA and glutamate, respectively, in region X with our behavioural variable of interest. As we detail in our new paragraph, these partial correlations can therefore be thought of much like a post hoc tests, asking whether GABA or glutamate

(or both) were driving the main effect of E/I balance in X. Thus, we did not indiscriminately run partial correlations on both GABA and glutamate in all regions, but only for those where there was a significant main effect of E/I balance.

Furthermore, we'd like to argue that the fact that we measured several brain areas, not just the two primary target areas (vmPFC and dACC) should indeed be seen as a particular strength of our study (as has also been appreciated by reviewer #1), not as a potential criticism. While we hope it is evident why dACC and vmPFC had been our primary areas of interest, we reasoned there might also be a contribution of dlPFC, at least to the patch-leaving stage and we also included the bilateral motor cortex as a control area, which we expected to show either no involvement, or, if anything, only with non-specific factors such as general response speed. We hope the reviewer agrees that the ability to make a statement on the role of E/I balance in two specific target areas, whilst controlling for potential influences of three other regions is more powerful than the conclusions obtained from a study that would have only measured the two target areas in the first place.

Finally, just to be sure, it sounds as if the reviewer assumed that we had partialled out the residuals for each region. If so, we'd like to clarify, here and in the manuscript, that this is not the case. We computed partial correlation by regressing out the main effect of all other potential contributors. As an example: For the correlation between vmPFC GABA and % correct choices, we regressed the effect of (1) glutamate from all five regions and (2) GABA from all regions except vmPFC out of both % correct choices *and* vmPFC GABA. We'd like to emphasize that this approach is highly conservative, as it removes any variance that cannot unambiguously be assigned to vmPFC GABA. Because we measured several brain regions, this furthermore allows us to remove any potential contributions from other brain regions that might otherwise be attributed to vmPFC GABA – and which could not be controlled for in studies that would not acquire these data in the first place.

1c) Multiple comparisons: *"Moreover, by running this analysis separately for each individual variable the authors need to correct for multiple comparisons."*

We would like to point out that the different statistical tests addressed different hypotheses. For both patch-leaving and value-based choice, we had one key test each: for patch-leaving, we tested for the effect of (dACC) E/I balance on PLA. For value-guided choice, we tested for the effect of E/I balance on % correct choice. This corresponds to the two hypotheses (1) that E/I balance in dACC (but not in the other four regions) is related to patch-leaving decisions and (2) that E/I balance in vmPFC (but not in the other four regions) is related to value-guided decision making. There is thus no need for multiple comparison correction, as

these analyses test two different things following directly from the two key hypothesis of the manuscript. To provide a more detailed picture, we then followed up these primary effects by investigating potential contributors. Again, this was strongly hypothesis-driven. For example, take the relationship between vmPFC E/I balance and effect of value difference on response speed. This is a prediction that emerges directly from biophysical models based on competition by mutual inhibition, showing that increasing the level of inhibition slows down the recurrent dynamics, therefore allowing for more evidence to be accumulated (noise averaging out over time) at the expense of slower response speed. In response both to your comment and to reviewer #1 (point 2a), we have added a section that describes this hierarchical approach in more detail, which we also reproduce below in blue for convenience:

Introduction:

We hypothesised that patch-leaving behaviour is guided by the balance between cortical excitation and inhibition (E/I balance) in dACC. In contrast, we expected that value-guided decision making is governed by E/I balance in vmPFC. Healthy human participants performed a novel decision-making task (Figure 1A) combining patch-leaving and value-based decision-making. We measured GABA and glutamate concentrations using magnetic resonance spectroscopy (MRS) at 7T in five cortical areas of interest: vmPFC, dACC, dorsolateral prefrontal cortex (dlPFC), and bilateral primary motor cortex (M1). Specifically, we predicted that interindividual differences in how costs and patch values influence behaviour relates to variations in E/I balance in dACC³⁻⁵ over and above the effects of all other voxels of interest. Further, we predicted that decision performance during value-guided choice would depend on vmPFC E/I balance^{6,7}. Additionally, models based on competition by mutual inhibition predict that the speed at which a decision unfolds is driven by the available evidence, and the rate of this evidence accumulation is again crucially dependent upon E/I balance⁶. We therefore further predicted that the effect of the key decision variables on response times would also be related to E/I balance in dACC and vmPFC, respectively.⁸

We report contributions of E/I balance that were dissociable as a function of decision type and cortical area. Patch leaving behaviour was related to E/I balance in dACC, but not in any of the other regions investigated. In contrast, value-guided decision making was related to E/I balance in vmPFC, but not in any of the other cortical areas.

Results:

We used multiple linear regression to test our hypotheses. In order to limit the number of statistical comparisons, we proceeded along the following hierarchy: Firstly, we only tested behavioural variables which we hypothesized to relate to E/I balance (see above). Secondly, using a general linear model (GLM), we first projected E/I balance (ratio between glutamate

and GABA) from all five regions of interest onto main behavioural parameters from the patch-leaving and value-guided choice phase. All of these analyses were performed exclusively using the design matrix containing E/I balance from all five regions. Thirdly, if, and only if, this GLM yielded a significant effect for one brain region, we followed this up by asking whether glutamate, GABA, or both within that specific region contributed to this effect of E/I balance. To this end, we then computed partial correlations, regressing out the effects of all other factors than the one currently of interest (see methods). These partial correlations can therefore be thought of as post hoc test, further investigating the individual contributions of GABA and glutamate to a main effect of E/I balance (if present).

Methods:

For each behavioural analysis, we obtained a measure describing the individual influence of key value parameters on behaviour ("decision variable"). To relate decision variables to cortical E/I balance, we used multiple linear regression. In order to limit the number of comparisons, we used the following hierarchy of testing: Firstly, we only tested those decision variables for their relationship with cortical neurochemistry (a) for which we had an a priori hypothesis and (b) that had a significant effect on behaviour. Secondly, using a general linear model (GLM), we first projected E/I balance (ratio between glutamate and GABA) from all five regions of interest against main behavioural parameters from the patch-leaving and value-guided choice phase. All of these analyses were performed exclusively using the design matrix containing E/I balance from all five regions. Thirdly, if, and only if, this GLM yielded a significant effect for one brain region, we followed this up by asking whether this effect of E/I balance was contributed to by glutamate, GABA, or both within that specific region. To this end, we then computed partial correlations, regressing out the effects of all other factors than the one currently of interest (see below). Fourthly, when there was an effect of E/I balance on a decision variable of interest, we followed this up by analysing in more detail what aspects of a decision variable were related to E/I balance.

We analysed the following decision variables: for the patch leaving phase, we considered the patch leaving advantage as the key measure of interest that describes how participants balance the expected advantage of leaving against the travel cost. For the value-guided choice phase, overall decision performance, measured as % correct choices (choices of the option with higher expected value) was the primary measure of interest. The latter relationship was investigated in further detail by testing to what extent the relationship between E/I balance and % correct was driven by the individual-specific underweighting of reward magnitudes as captured by the model-derived parameters α and γ . Finally, we assessed how E/I balance was related to the effect of relevant reward information on

reaction times. Here, for the patch-leaving phase, the effect of cost on response speed was our key parameter of interest, whereas for the value-guide choice phase, we focussed on the effect of value difference on reaction times. As a control, we also included overall response times, independent of any task parameters (Supplementary Table 4 and 5 for overview).

All of the E/I analyses were performed exclusively using the design matrix containing balances from all five regions. The ratios of glutamate to GABA in all voxels of interest were entered as regressors in a design matrix (along with a constant term) to predict the contributions of E/I balance onto each decision variable (one linear model per decision variable, all variables normalized). In addition to t- and p- values we report the 95 % confidence intervals for each significant ($p < 0.05$) regression coefficient. Only if a significant influence of E/I balance in a specific region was identified, we computed partial correlations. These partial correlations were first computed for E/I balance in the target region (the one showing a significant effect in the main GLM) by orthogonalizing (removing the effect of all other E/I balances) both from the decision variable and from E/I balance in the target region. These partial correlations (Pearson correlation between the residuals of decision variable and E/I balance) are what is shown in 2 C, D and 3 B, C, D. To further investigate whether a main effect of E/I balance in one region was driven by GABA or glutamate (or both), we computed further partial correlations on the orthogonal contribution of GABA and glutamate. To do so, we orthogonalized both the decision variable and the neurotransmitter of interest (GABA or glutamate, respectively) in the target area with respect to both the respective other neurotransmitter in the same region and both GABA and glutamate in all other voxels. As an example, if the GLM detected a main effect of E/I in dACC on patch-leaving advantage, and we wanted to compute the orthogonal contribution of dACC GABA to this effect, then we removed the effect of dACC glutamate and the effects of both GABA and glutamate in the other four voxels from both the patch-leaving advantage and from dACC GABA. Again, we computed Pearson correlations between the residuals of the decision variable and the residuals of GABA or glutamate, respectively. In the analysis of value-guided decisions, we also controlled for the amount of no brainer trials (mean \pm SEM: 38.52 % \pm 0.01) as a predictor variable of no interest. We report the correlation results along with their respective p-values and 95 % confidence intervals for each correlation coefficient. We report the respective r- and p- values for each correlation as well as the 95 % confidence interval for each regression coefficient.

2) "A much more straightforward approach would be to run a regression analysis for each voxel (i.e. brain region) where the coefficients for each decision variable are simultaneously entered as predictors."

We thank the reviewer for this suggestion. Our approach had been to ask where in the brain variance in E/I balance explains variance in a behaviour of interest. The approach suggested by the reviewer would instead ask what combination of variance in behavioural parameters explains variance in one cortical region's E/I balance. However, we agree that this is an interesting alternative approach since it would also take into account potential correlations between the different behavioural variables (which we now provide in Supplementary Table 1 and 2). We have added these analyses as Supplementary Analysis 3 and reproduce them below for the reviewer's convenience. Please note however that, unlike our main analysis approach, this analysis does not account for correlation in E/I balance across regions. We would therefore see them as complementary to our previous analyses.

3) Simultaneous regression of all behavioural parameters of interest against E/I balance in dACC and vmPFC

Some of our dependent variables may be correlated with each other across participants. This is expected since some of the tests investigate parameters that we assume to be driven by a shared underlying mechanism¹⁴. For instance, consider the case for % correct choices on one hand and the effect of value difference on RT on the other. As can be seen from Supplementary Table 2, there is a negative correlation between these two variables, indicating that the (negative) effect of value difference on RT is most pronounced in participants with high percentage of correct choices. This, however, is exactly what would be mechanistically predicted from models using competition via mutual inhibition: Slowing the decision in the face of a lot of noise (a difficult trial with low value difference) allows for the choice to be dominated by the available evidence, while averaging out (neural) noise over time. To assess the orthogonal contributions of all the different behavioural parameters across both the patch-leaving and value-guided choice phase, we therefore included all of the parameters of interest from both phases (Supplementary Table 1 and 2) into one single regression model and now used either dACC or vmPFC E/I balance as the dependent variable. We still find a significant effect of patch leaving advantage on dACC E/I balance ($t_{22} = 3.105$, $p = 0.006$, $CI_{95} = [0.191 - 0.973]$) but no significant effect of any other variable of interest (all $p > 0.128$). When regressing the same design matrix against vmPFC E/I balance, we find no significant effect of any behavioural parameter (all $p > 0.159$).

As outlined in this above paragraph, one should not necessarily expect all dependent behavioural variables to be entirely uncorrelated as they may share underlying mechanisms. For instance, consider (value-guided choice) the case for % correct on one hand and the effect of value difference on RT. As you can see from our Supplementary Table 2, there is a negative correlation between these two variables, indicating that the (negative) effect of value difference on RT is most pronounced in participants with high percentage of correct choices. This, however, is exactly what would be mechanistically predicted from models using competition via mutual inhibition: Slowing the decision in the face of a lot of noise (a difficult trial with low value difference) allows for the choice to be dominated by the available evidence, while averaging out (neural) noise over time. It is still of high relevance to show that both variables are related to one another.

Reviewer 3

"[...] Overall, the research question is clearly defined, the experimental design is well conceived and conducted with appropriate care for details. The manuscript is well written. The discussion presents a very knowledgeable and physiologically sound interpretation of the findings and integrates a lot of seminal animal data on E/I balance."

We thank the reviewer for the positive and careful assessment of our work. Furthermore, we are particularly grateful for the reviewer's very specific suggestions for further analyses, to which we respond in detail below.

1) "However, the computational modeling left me a bit underimpressed. The model simply replicates, what has been shown in the model-free analyses and does not elucidate the cognitive processes underlying the foraging decision in the task any further. In its current version, I would recommend this paper to a more specialized journal rather than Nature Communications with its very broad readership. My recommendation for the paper would be a more sophisticated modeling approach to overcome the "cognitive sparseness" of the current version of the paper."

We'd like to start our response to this comment by pointing out that this manuscript was never meant to be primarily a modelling study, nor was modelling even meant to play a central role in this manuscript. Instead, the primary aim of this study was to relate cortical neurochemistry to two kinds of decision processes. Whether these decision processes are quantified using model-free approaches (e.g. regression-based analyses) or by fitting algorithmic models is, in our view, secondary. In our opinion, important insights into brain-

behaviour relationships can be gained even when modelling is not at the centre-stage, and we hope the reviewer would agree with this.

This said, we are very grateful for the reviewer's comment, because we agree that further elaborating the modelling approaches will strongly improve the quality of our study. In particular, following the approach recently described by Farashahi and colleagues (2019)⁹ in humans and non-human primates, we have now incorporated twelve different models. These models feature different ways of how reward information (probabilities and magnitudes) is combined (additive, multiplicative, or a weighted combination) and whether participants exhibit systematic distortions in the weighting of reward information. The models we had used in our previous submission are included in the set of models. Please see Supplementary Table 6 for details of all models we now present. In our view, these new computational modelling results provide important novel information in addition to the regression-based results. Firstly, it shows that the data is best explained assuming both multiplicative value integration and systematic distortions in the weighting of both parameters. These findings will be of high interest to a large community of researches concerned with the question of how value is constructed from individual attributes like probability and magnitude. Secondly, we can show that the systematic distortions in reward weighting (i) are related to vmPFC E/I balance and (ii) potentially mediate the relationship between value-guided choice accuracy and vmPFC E/I balance, thereby relating interindividual variation in value construction to neurotransmitter balance.

2) "The present finding on the effect of E/I balance on RTs already hints at a modulation of the speed-accuracy trade-off. Therefore, I would recommend a modeling approach along the lines of drift diffusion modeling, which is able to capture this trade-off really well. I think this approach could potentially uncover the intriguing finding that the E/I balance in a specific region may modulate the boundary separation or drift rate in these models, which would make it cognitively more interesting and meaningful for a broader audience. In addition, probability, magnitude and expected value may play a modulatory role in the framework, which could provide an informative link to the model-free results."

We want to thank the reviewer for suggesting this. Firstly, we agree that analyzing our data using drift diffusion models (DDM) will link our work to that of a large research community studying evidence accumulation in this framework. Secondly, as the reviewer already hints at, the assumed biophysical mechanism (competition by mutual inhibition in recurrent networks exhibiting attractor dynamics) can indeed be cast as biological implementation of a

DDM (coarsely speaking). We have therefore fitted hierarchical Bayesian DDM to the data both from the patch-leaving and the value-guided choice phase. We should point out that, unlike for value-guided decision making, to our knowledge no mechanistic models exist that can capture accumulation of evidence across trials, i.e. over seconds to tens of seconds (as is the case for the patch-leaving phase).

We have fitted hierarchical DDMs separately to the data from the patch-leaving and value-guided choice stage. In brief, we find that, for the patch-leaving phase, a drift diffusion model with free parameters boundary, drift rate, non-decision time, bias towards one boundary, trial-to-trial variability in non-decision time and trial-to-trial variability in bias (a , v , ndt , z , st and sz , respectively) fit the data best. When assessing the relationship between model parameters and E/I balance, we find a marginally significant effect of dACC E/I balance on drift rate ($t_{23} = 2.011$, $p = 0.056$, $CI_{95} = [-0.012 - 0.837]$, $r = 0.387$, $p = 0.038$, $CI_{95} = [0.023 - 0.660]$). This indicates that participants with a greater dACC E/I balance show a higher drift towards stay decisions, which confirms our regression-based findings. There were no significant effects in any other region of interest (all $p > 0.629$) nor with decision boundary (all $p > 0.269$). For the value-guided choice phase, we find that a DDM incorporating a regression model on drift rate fits the data best. While we don't find any main effect of vmPFC E/I on drift rate, we find a interaction effect between vmPFC E/I and the effects of value difference on drift rate. Again, this pattern of results exactly matches our findings obtained from the regression analyses (Effect of value difference on response speed and the relationship of this effect with vmPFC E/I) and points towards an influence of vmPFC E/I balance onto the speed of value integration.

We have incorporated the analysis in more detail below. However, while we agree that adopting a DDM framework to study our data might help to bridge different fields (in particular value-guided vs perceptual decision making), we do not see any clear added value of the DDM over and above our regression-based analyses. Indeed, we would argue that there is a direct correspondence: the above relationship between dACC E/I balance and drift rate is in our view analogous to that with patch-leaving advantage: with higher levels of dACC glutamate relative to GABA, participants require more evidence in favour of leaving to trigger a patch exit (model-free: patch-leaving advantage) and they show a stronger drift towards the "stay" decision (DDM: drift rate). Similarly, in the value-guided choice phase, participants with higher vmPFC levels of GABA relative to glutamate slow down their decisions more strongly on trials with low value difference (regression: effect of value difference on RT) and they show a slower drift to the boundary (DDM: effect of value difference on drift rate). Therefore, we would argue that these DDM results, while consistent

with our model-free analyses, are also somewhat redundant. Since regressions are merely an algebraic projection of the data and, unlike DDM, do not rely on fitting a high-dimensional parameter space, we would argue that our model-free analyses are more parsimonious. Further to this, including the DDM would increase the number of statistical tests performed (for what we think is at large testing the same thing twice) and we would therefore prefer not to present the DDM data at all. Further to this, we have included E/I balances directly in the hierarchical models as a cross-level interaction. While this helps us to assess the effects of E/I balance within the same Bayesian framework as the regression coefficients, it also makes our models overly complex and leads to convergence problems in regression models explaining variance in the decision boundary. We did this because we wanted to avoid correlating individual slope estimates with E/I balances since they might be shrunk towards the group mean due to the hierarchical design of the model (whereas the between – subjects variable E/I is not). However, for the analysis of patch leaving data, the model without including dACC E/I balance fitted the data best. We have therefore run an exploratory analysis to relate individual parameters to E/I balance after model fitting. If the reviewer disagrees with our reasoning and judges these findings to be useful, we would of course be happy to include them, preferably in the supplements.

Drift Diffusion Modelling of Choice Data

To obtain a formal characterization of the process of evidence accumulation across trials, we fitted a hierarchical drift diffusion model (DDM)¹⁶ In brief, DDM assume that choices between two alternatives depend upon accumulation of noisy evidence until a decision threshold is reached. The model thereby not only predicts choice probabilities but also response time (RT) distributions. The predicted choice probabilities and RTs critically depend upon three free parameters. First, the decision boundary a determines how much evidence needs to be accumulated. Second, the drift rate v captures the speed at which the evidence accumulation process approaches either boundary¹⁶. Third, RT is assumed not to solely depend on the choice process itself, but also on other non-decisional processes like stimulus perception and the execution of a motor response, which is reflected in the non-decision time (ndT)¹⁷. To account for across-trial variations¹⁸, we also tested the effects of variability in ndt (st) and v (sv).

We used the Bayesian hierarchical drift diffusion modeling toolbox¹⁶ to infer latent variables underlying response time distributions of patch leaving trials and correct vs. incorrect value - guided choices. The estimation of individual parameters is hierarchical since they are not assumed to be independent of one another but drawn from an underlying group

distribution¹⁶. We estimated drift rate, boundary separation and non-decision time individually, but across-trial variability in drift rate and non-decision time on a group level^{19,20}. For model comparisons, we included additional effects of bias towards one decision boundary (z) and variations in bias (sz) in patch leaving trials and linear regression models assessing the effect of reward information onto free DDM parameters. For regressions, all continuous variables were z-scored per participant before estimating regression coefficients. Cost levels were z-scored on a group level. Since our task does not involve a maximum response time, we excluded all trials with response times below 0.3 or above 4 seconds before model fitting. Additionally, we specified 5 % of responses to be contaminants. The toolbox uses Markov-Chain Monte Carlo sampling for a Bayesian approximation of the posterior distribution of each model parameter. For every model, we ran thirty separate Markov chains and report parameter estimates and posterior distributions of a concatenated model across all chains. We generated 5000 samples for every chain and discarded one half of all samples as burn-in. Every third sample was discarded for thinning, thereby reducing autocorrelations in the chains¹⁹. To assess model convergence, we inspected the sampled posterior traces, their autocorrelation and the Gelman-Rubin \hat{R} statistics, which compares between and within chain variance^{16,21}. \hat{R} for a group level parameter with a distance of > 0.02 from one were defined as non-converged models¹⁹. To compare between models, we used the Deviance Information Criterion (DIC) where a lower DIC points towards a better fit. Based on previous findings^{8,22-25}, we predicted a relationship between E/I balance and the drift rate v and decision boundary a .

For the patch leaving phase we find that a drift diffusion model with a , v , ndt , z , st and sz fit the data best. When we assess the effects for individual model parameters for their relationship with E/I balance²⁶, we find a significant effect of dACC E/I balance on drift rate ($t_{23} = 2.011$, $p = 0.056$, $CI_{95} = [-0.012 - 0.837]$, $r = 0.387$, $p = 0.038$, $CI_{95} = [0.023 - 0.660]$). This indicates that participants with a greater dACC E/I balance show a higher drift towards stay decisions and confirms our model free findings. There were no significant effects in any other region of interest (all $p > 0.629$) nor with decision boundary (all $p > 0.269$). We included E/I balances directly in the model rather than correlating E/I balances with individual slopes (after model fitting) since the latter might be biased towards the group mean. However, since the standard DDM without incorporating E/I balance fitted the patch-leaving data best, we ran the exploratory analysis reported above.

For the value guided choice phase, we find that a DDM incorporating a regression model on drift rate fits the data best. While we do not find any main effect of vmPFC E/I on drift rate (highest posterior density interval (HPDI): $[-0.279 - 0.052]$), we find an interaction

effect between vmPFC E/I and the effects of value difference on drift rate (HPDI: [0.032 – 0.124], <0.001 % of the posterior distribution below zero). Additionally, we find an overall greater drift rate with higher value difference between options (HPDI: [0.515 – 0.627], <0.001% of distribution below zero) and in no brainer trials (HPDI: [1.843 – 2.117], <0.001% of distribution below zero). This pattern of results matches our findings obtained in regression analyses and again points towards an influence of vmPFC E/I balance onto the speed of value integration.

Table 1: Overview of DDM models: Overview of HDDM model specifications.

Free Parameters	Linear Model	DIC	Gelman Rubin
Patch Leaving Phase			
a, v, ndt, sv, st		3145.12	yes
a, v, ndt, z, sz, st		3104.46	yes
a, v, ndt, z, st, sz	$v \sim 1 + \text{costs} + \text{dacc} + \text{costs}:\text{dacc}$	4205.75	yes
a, v, ndt, z, st, sz	$a \sim 1 + \text{costs} + \text{dacc} + \text{costs}:\text{dacc}$	4189.73	no
Value guided choice phase			
a, v, ndt, sv, st		15112.98	yes
a, v, ndt, sv, st	$v \sim 1 + \text{valdiff} + \text{NB} + \text{vmpfc} + \text{valdiff}:\text{vmpfc}$	11818.15	yes
a, v, ndt, sv, st	$a \sim 1 + \text{valdiff} + \text{NB} + \text{vmpfc} + \text{valdiff}:\text{vmpfc}$	14290.62	no

Minor Comments

M1) "Throughout the paper I was wondering, if the E/I balance in dACC was correlated with the one in vmPFC and whether this could explain the differential effects seen for the 1st and 2nd decision. This would be useful to include in a revised version of the manuscript."

This is an important point. Importantly, by putting the E/I balances from all five cortical regions into one GLM, we allow them to compete for variance in the dependent variable, thus revealing only the orthogonal contribution of the five regions. Furthermore, by running partial correlations we have followed the most conservative approach by explicitly removing all

variance that cannot uniquely be attributed to the neurotransmitter/cortical area of interest. In addition, we now report the correlation of E/I balances between the five regions in our new Supplementary Table 3.

M2) "I was wondering why the differential findings for GABA and Glutamate in dACC on logRT did not make it into a figure."

Our apologies, we are of course happy to provide the figure the reviewer is asking for, but we must admit that we are not sure what the reviewer is asking for specifically. Does this remark refer to the patch leaving phase and the partial correlations between GABA/glutamate and the cost effect on logRT? Or is it the value-guided phase and the partial correlations between GABA/glutamate and overall logRT? In both cases, we found a main effect of dACC E/I balance (which we now show in Figure 2D and Supplementary Figure 2B) and the partial correlations in the second case showed only an effect of GABA, but in both cases no significant contributions of glutamate. Please let us know what visualization you are referring to and we are happy to include it.

References

1. Palminteri, S., Wyart, V. & Koechlin, E. The Importance of Falsification in Computational Cognitive Modeling. *Trends in cognitive sciences* **21**, 425–433 (2017).
2. Wilson, R.C. & Collins, A.G. Ten simple rules for the computational modeling of behavioral data. *eLife* **8** (2019).
3. Kolling, N. *et al.* Value, search, persistence and model updating in anterior cingulate cortex. *Nat. Neurosci.* **19**, 1280 (2016).
4. Kvitsiani, D. *et al.* Distinct behavioural and network correlates of two interneuron types in prefrontal cortex. *Nature* **498**, 363–366 (2013).
5. Hayden, B.Y., Pearson, J.M. & Platt, M.L. Neuronal basis of sequential foraging decisions in a patchy environment. *Nat. Neurosci.* **14**, 933–939 (2011).

6. Jocham, G., Hunt, L.T., Near, J. & Behrens, T.E.J. A mechanism for value-guided choice based on the excitation-inhibition balance in prefrontal cortex. *Nat. Neurosci.* **15**, 960–961 (2012).
7. Hämmerer, D., Bonaiuto, J., Klein-Flügge, M., Bikson, M. & Bestmann, S. Selective alteration of human value decisions with medial frontal tDCS is predicted by changes in attractor dynamics. *Scientific reports* **6**, 25160 (2016).
8. Standage, D. & Paré, M. Persistent storage capability impairs decision making in a biophysical network model. *Neural networks : the official journal of the International Neural Network Society* **24**, 1062–1073 (2011).
9. Shiva Farashahi, Christopher H. Donahue, Benjamin Y. Hayden, Daeyeol Lee & Alireza Soltani. Flexible combination of reward information across primates. *Nat Hum Behav* **3**, 1215–1224 (2019).
10. Greenhouse, I., King, M., Noah, S., Maddock, R.J. & Ivry, R.B. Individual differences in resting corticospinal excitability are correlated with reaction time and GABA content in motor cortex. *J. Neurosci.* **37**, 2686–2696 (2017).
11. Houtepen, L.C. *et al.* Acute stress effects on GABA and glutamate levels in the prefrontal cortex: A 7T 1H magnetic resonance spectroscopy study. *NeuroImage. Clinical* **14**, 195–200 (2017).
12. Near, J., Ho, Y.-C.L., Sandberg, K., Kumaragamage, C. & Blicher, J.U. Long-term reproducibility of GABA magnetic resonance spectroscopy. *NeuroImage* **99**, 191–196 (2014).
13. Talsma, L., van Loon, A., Scholte, H.S. & Slagter, H.A. State or trait? MRS-measured GABA and Glutamate concentrations are not modulated by task demand and do not robustly predict task performance. *bioRxiv* **154** (2019).
14. Wang, X.-J. Probabilistic decision making by slow reverberation in cortical circuits. *Neuron* **36**, 955–968 (2002).
15. Hunt, L.T. *et al.* Mechanisms underlying cortical activity during value-guided choice. *Nature neuroscience* **15**, 470 (2012).
16. Wiecki, T.V., Sofer, I. & Frank, M.J. HDDM: hierarchical bayesian estimation of the drift-diffusion model in python. *Frontiers in neuroinformatics* **7**, 14 (2013).
17. Palmer, J., Huk, A.C. & Shadlen, M.N. The effect of stimulus strength on the speed and accuracy of a perceptual decision. *Journal of vision* **5**, 1 (2005).

18. Boehm, U. *et al.* Estimating across-trial variability parameters of the Diffusion Decision Model: Expert advice and recommendations. *Journal of Mathematical Psychology* **87**, 46–75 (2018).
19. Urai, A.E., Gee, J.W. de, Tsetsos, K. & Donner, T.H. Choice history biases subsequent evidence accumulation. *eLife* **8** (2019).
20. Ratcliff, R. & Childers, R. Individual Differences and Fitting Methods for the Two-Choice Diffusion Model of Decision Making. *Decision (Washington, D.C.)* **2015** (2015).
21. Gelman, A. & Rubin, D.B. Inference from iterative simulation using multiple sequences. *Statistical science* **7**, 457–472 (1992).
22. Fouragnan, E.F. *et al.* The macaque anterior cingulate cortex translates counterfactual choice value into actual behavioral change. *Nat. Neurosci.* **22** (2019).
23. Khalighinejad, N. *et al.* A Basal Forebrain-Cingulate Circuit in Macaques Decides It Is Time to Act. *Neuron* **105**, 370-384.e8 (2020).
24. Brockett, A.T., Tennyson, S.S., deBettencourt, C.A., Gaye, F. & Roesch, M.R. Anterior cingulate cortex is necessary for adaptation of action plans. *Proceedings of the National Academy of Sciences of the United States of America* **117**, 6196–6204 (2020).
25. Wang, X.-J. Neural dynamics and circuit mechanisms of decision-making. *Current opinion in neurobiology* **22**, 1039–1046 (2012).
26. Katahira, K. How hierarchical models improve point estimates of model parameters at the individual level. *Journal of Mathematical Psychology* **73**, 37–58 (2016).

REVIEWER COMMENTS

Reviewer #1 (Remarks to the Author):

The authors have fully addressed all of my comments. In particular, it is worth highlighting that the authors have made a great effort to increase the thoroughness and exhaustiveness of the computational modelling, which will make the paper even more interesting to many researchers in the field.

Reviewer #2 (Remarks to the Author):

I appreciate the effort the authors have gone to in clarifying the statistical techniques used. I agree that the tests were appropriate, as they are attempting to determine if the coefficients relating the E/I balance to the behavioral measures differ from zero for each region holding constant the potential influence of the other regions. Nonetheless, the patch leaving advantage correlation with the E/I balance appears to be completely driven by two outliers (Figure 2B). What is being plotted in the Figure? The raw E/I measure or the residuals from the GLM? If the latter I am concerned because it would appear that the entire correlation hinges on these two points.

It is also unclear why the t-statistic and r-values aren't associated with the same p-value for the reported E/I results, For example: "Regressing E/I balance against patch leaving advantage revealed a significant influence of E/I balance in dACC ($t_{23}=2.643$, $p=0.015$, $CI_{95}=[0.111-0.908]$; $r=0.483$, $p=0.008$, $CI_{95}=[0.141-0.722]$, Figure 2C) but not in any other region of interest (all $p>0.199$)." Based on the reported t-statistic the correlation should be $0.2087 \left(\frac{(2.463.^2)}{(2.463.^2+23)} \right)$. So how do they arrive at the reported correlation value? This needs further clarification.

Reviewer #3 (Remarks to the Author):

I want to thank the authors for putting a lot of additional work to fully addressing my questions. I am supportive of a publication now.

As for the two remaining points:

I would recommend to include the DDM modeling in the supplement and I think this is useful additional information for those readers who want to dig in a bit deeper.

Furthermore, I am content with including the GABA effects as supplemental figures.

Congratulations on an interesting and well-written paper.

Jan Gläscher

We are thankful to the reviewers for their overall very positive evaluation of our revised manuscript. While both reviewers 1 and 3 are supportive of publication of the manuscript in its present form (subject to some changes suggested by reviewer 3), reviewer 2 has some remaining concerns. We would therefore like to begin by briefly addressing reviewers 1 and 3 before turning to reviewer 2 in more detail. We reproduce each of the reviewers' comments below in italics before our response to each point. Changes made to the manuscript are written in blue, both here and in the manuscript.

Reviewer #1:

The authors have fully addressed all of my comments. In particular, it is worth highlighting that the authors have made a great effort to increase the thoroughness and exhaustiveness of the computational modelling, which will make the paper even more interesting to many researchers in the field.

We want to again thank reviewer 1 for their helpful and supportive comments that have led to a great improvement over our initial submission – and also for the positive final evaluation.

Reviewer #3:

I want to thank the authors for putting a lot of additional work to fully addressing my questions. I am supportive of a publication now. As for the two remaining points: I would recommend to include the DDM modeling in the supplement and I think this is useful additional information for those readers who want to dig in a bit deeper. Furthermore, I am content with including the GABA effects as supplemental figures. Congratulations on an interesting and well-written paper.

Jan Gläscher

Dear Dr Gläscher,

Thank you again for your detailed comments and for your supportive feedback on our revised manuscript. Following your recommendation, we have now included the DDM results in the supplementary material – and we have likewise included a supplementary figure (Figure S2C) for the GABA effects. Again, thank you for a constructive and helpful review.

Reviewer #2:

1) I appreciate the effort the authors have gone to in clarifying the statistical techniques used. I agree that the tests were appropriate, as they are attempting to determine if the coefficients relating the E/I balance to the behavioral measures differ from zero for each region holding

constant the potential influence of the other regions. Nonetheless, the patch leaving advantage correlation with the E/I balance appears to be completely driven by two outliers (Figure 2B). What is being plotted in the Figure? The raw E/I measure or the residuals from the GLM? If the latter I am concerned because it would appear that the entire correlation hinges on these two points.

The reviewer is correct, figure 2C indeed shows the residuals from the GLM. There are indeed three values for E/I larger than 1.5 - which is likely difficult to see because of the two overlying dots at the highest E/I values. We would like to apologize for this confusion and we now provide the figures in higher resolution in the present submission.

To assess whether our correlation is driven by potential outliers, we first tested whether any of the data points in question is more than three standard deviations from the mean. This is not the case for any data point. Nevertheless, we reanalyzed our data using robust regression analysis and estimated the Cook's distance for all observations within the linear model. In a robust regression model with bisquare weighting, the weight given to each observation depends on how far the point is from the fitted regression line. It thereby minimizes the effect of outliers on the estimated regression slope. Using this approach, we still find a significant influence of dACC E/I balance on the patch leaving advantage ($t_{23} = 2.488$, $p = 0.021$) with no significant effect of any other region of interest (all $p > 0.218$). Furthermore, we estimated the Cook's distance, combining each observation's leverage and residuals to identify outliers. We excluded all observations with a Cook's distance three times greater than the mean of all observed distances. On these grounds we excluded two observations (observation 5 and observation 12). When we fitted an GLM again without these observations, we still find a significant effect of dACC E/I balance on PLA ($t_{21} = 2.301$, $p = 0.032$, $CI_{95} = [0.059 - 1.175]$; $r = 0.449$, $p = 0.019$, $CI_{95} = [0.083 - 0.708]$). On these grounds, we feel confident that the relationship between dACC E/I balance and patch leaving advantage shown in Figure 2C is not driven by outliers.

2) It is also unclear why the t-statistic and r-values aren't associated with the same p-value for the reported E/I results, For example: "Regressing E/I balance against patch leaving advantage revealed a significant influence of E/I balance in dACC ($t_{23}=2.643$, $p=0.015$, $CI_{95}=[0.111-0.908]$; $r= 0.483$, $p= 0.008$, $CI_{95} = [0.141-0.722]$, Figure 2C)but not in any other region of interest (all $p>0.199$)." Based on the reported t-statistic the correlation should be $0.2087 ((2.463.^2)/((2.463.^2)+23))$. So how do they arrive at the reported correlation value? This needs further clarification.

We believe there may be a misunderstanding, our apologies for not being clear enough in our previous response. The different p-values ($p = 0.015$ and $p = 0.008$) arise from

the fact that we first used a general linear model with six predictors (five E/I balances plus a constant term). The first p-value ($p = 0.015$) therefore represents the p-value for the test that dACC E/I balance has an effect bigger than zero on patch leaving advantage, given the other predictors in the model. We followed this up by computing partial correlations - this is also what we used for illustration purposes (in this case, Figure 2C). Here we orthogonalized both dACC E/I balance and the behaviour of interest (patch leaving advantage) with respect to E/I balance in all other regions of interest by removing their contributions from both variables - we computed partial correlations between the resulting residuals. The second p-value ($p = 0.008$) is the p-value for this partial correlation. The p-values are different since the error degrees of freedom in the correlation of the residuals are 27 whereas they are 23 in the regression model.

Regarding the reviewer's question about the t-statistic and the reported correlation, we would kindly like to ask whether it is possible that the reviewer missed the square root for deriving the correlation coefficient from the t-statistic? Based on the t-score, we would arrive at a correlation of $\sqrt{(2.643.^2)/(2.643.^2+23)} = 0.483$, which is what we report.

Further to these two comments, the editor also alerted us that reviewer #2 had, in their private remarks to the editor, reiterated the need to correct for multiple comparisons. This is a concern that had been brought up in the previous round, and we had responded in detail to this (in part also by elaborating our description following the structure suggested by reviewer #1). We would have been keen to respond to the remaining concerns reviewer #2 had. However, since we do not know which aspect(s) of our response the reviewer found not fully convincing, we can therefore only briefly reiterate what we believe might be the key points:

- Firstly, different statistical tests were performed to test different things (e.g. hypothesis 1: patch-leaving is related to dACC E/I balance, hypothesis 2: value-guided choice is related to vmPFC E/I balance)
- Secondly, and this reasoning also seems to be supported by reviewer #1 in their previous review, there is no need to correct for the number of brain areas because there were clear a priori hypotheses with resulting pre-planned tests. Again, our goal was to estimate (i) the contribution of dACC E/I balance to patch-leaving and (ii) the contribution of vmPFC E/I balance to value-guided choice, both while accounting for the effect of E/I balance in the respective other four cortical regions. That these hypotheses were indeed set up a priori should be easily evident not only from the literature, but also from our own previous work (Jocham et al., 2012). Indeed, the experiment here and the hypotheses had been laid down in a grant proposal submitted to the German Research Foundation (Deutsche Forschungsgemeinschaft) as early as 2015. Thus, it should be clear that the respective other

voxels merely serve control purposes and correcting for number of brain areas would therefore, in our opinion, lead to an undue increase in the chance of type II error.

The editor also informed us that reviewer #2, again in their remarks to the editor, had shared that they remain concerned with our approach relating MRS to behaviour. We believe that we had responded to this concern in detail in our previous response to reviewers, and unfortunately it remains opaque to us precisely what aspects of our approach the reviewer deems inadequate. We are particularly surprised because, in their first sentences, the reviewer indeed does not give us the impression that there are remaining issues about our approach ("*I appreciate the effort the authors have gone to in clarifying the statistical techniques used. I agree that the tests were appropriate....*"). However, our understanding is that the editor will ask reviewers #1 and #3 to comment on what has been written so far. We would thus refrain from reiterating our reasoning at this stage but would of course be happy to address these if the reviewers point us to parts that require further clarification or modifications.

Finally, we performed the following minor modifications:

1) When we analysed the relationship between reaction times (RT) and E/I balance on a group level, we have previously used the log RT. However, since the analysis here was *across* participants (rather than *within*), the independent variable is the median RT (rather than trial-wise RT), which is a summary statistic and not prone to the same issues as trial-wise RT. Hence, we re-ran this analysis simply using each volunteer's median RT, without log-transformation. This resulted in very minimal changes in p-values and did not affect our inferences.

2) We have additionally added the following sentence to our methods section: "Because of our clearly defined a priori hypotheses regarding the role of dACC vs vmPFC, we did not apply correction for multiple comparison based on the number of brain regions tested in our regression models."

REVIEWERS' COMMENTS

Reviewer #1 (Remarks to the Author):

Reviewer #2

"I appreciate the effort the authors have gone to in clarifying the statistical techniques used. I agree that the tests were appropriate, as they are attempting to determine if the coefficients relating the E/I balance to the behavioral measures differ from zero for each region holding constant the potential influence of the other regions. Nonetheless, the patch leaving advantage correlation with the E/I balance appears to be completely driven by two outliers (Figure 2B). What is being plotted in the Figure? The raw E/I measure or the residuals from the GLM? If the latter I am concerned because it would appear that the entire correlation hinges on these two points.

It is also unclear why the t-statistic and r-values aren't associated with the same p-value for the reported E/I results, For example: "Regressing E/I balance against patch leaving advantage revealed a significant influence of E/I balance in dACC (t₂₃=2.643, p=0.015, CI₉₅ = [0.111–0.908]; r= 0.483, p= 0.008, CI₉₅ = [0.141–0.722], Figure 2C) but not in any other region of interest (all p>0.199)." Based on the reported t-statistic the correlation should be 0.2087 ($(2.463.^2)/(2.463.^2+23)$). So how do they arrive at the reported correlation value? This needs further clarification."

Opinion about reviewer comments:

Just to clarify, I think in figure 2 the reviewer refers to, probably plots C and D (not B).

This is the figure 2 the reviewer refers to, probably plots C and D (not B).

Point 1: Parametric vs. non-parametric correlation

On examining the figure more properly, I would partly agree with reviewer 2: from the figure it is a bit unclear whether the correlation should be parametric or non-parametric.

It seems that it would be pretty simple for the authors to address this question: examine whether parametric or non-parametric correlations should be used via standard methods and then if needed apply non-parametric correlation.

Point 2: relationship between reported t-values and correlation.

I think the discrepancy between the t-values and the r-values is that the t-values refer to the results of the regression, while the r-values refer to the subsequent partial correlations. It would be good for the authors to clarify, but I think this is not a point for concern.

Reviewer #2 (Remarks to the Author):

All of my concerns regarding the regression analyses results are resolved.

Reviewer #3 (Remarks to the Author):

I have carefully re-read the comments from reviewer #2 and the author's responses in both rounds of review. I find that the author's went out of their way to explain their statistical approach and gave very clear description of the analyses. The additional analyses using robust regression and Cook's distance to identify outliers are appropriate and underline the author's claims of an effect of dACC E/I balance. Because I also don't know what reviewer #2 communicated to the editors about his remaining concerns, I cannot evaluate, whether these have been addressed by the authors. But from the material that I have at my disposal, I would think that they responded to the comments to the best of their knowledge and abilities and would support a publication of the paper in NCOMMS.

We would like to thank all reviewers for the time and effort they put in our manuscript. We are delighted by the overall positive evaluation and address the two remaining concerns in detail below. We reproduce each of the reviewer's comments below in italics before our response to each point. Changes made to the manuscript are written in blue and highlighted with the track changes feature in Microsoft Word in the manuscript.

Reviewer #2

1) I appreciate the effort the authors have gone to in clarifying the statistical techniques used. I agree that the tests were appropriate, as they are attempting to determine if the coefficients relating the E/I balance to the behavioral measures differ from zero for each region holding constant the potential influence of the other regions. Nonetheless, the patch leaving advantage correlation with the E/I balance appears to be completely driven by two outliers (Figure 2B). What is being plotted in the Figure? The raw E/I measure or the residuals from the GLM? If the latter I am concerned because it would appear that the entire correlation hinges on these two points.

2) It is also unclear why the t-statistic and r-values aren't associated with the same p-value for the reported E/I results, For example: "Regressing E/I balance against patch leaving advantage revealed a significant influence of E/I balance in dACC ($t_{23}=2.643$, $p=0.015$, $CI_{95}= [0.111-0.908]$; $r= 0.483$, $p= 0.008$, $CI_{95} = [0.141-0.722]$, Figure 2C) but not in any other region of interest (all $p>0.199$)." Based on the reported t-statistic the correlation should be $0.2087 ((2.463.^2)/((2.463.^2)+23))$. So how do they arrive at the reported correlation value? This needs further clarification.

Opinion of Reviewer #1 about Reviewer #2 comments:

1) Just to clarify, I think in figure 2 the reviewer refers to, probably, plots C and D (not B).

Point 1: Parametric vs. non-parametric correlation

On examining the figure more properly, I would partly agree with reviewer 2: from the figure it is a bit unclear whether the correlation should be parametric or non-parametric. It seems that it would be pretty simple for the authors to address this question: examine whether parametric or non-parametric correlations should be used via standard methods and then if needed apply non-parametric correlation.

We want to thank reviewer 1 for providing their opinion on this point raised by reviewer 2. In the following we aim to clarify why we have reported parametric correlations, despite the three data points that appear further away from the remainder of the data in figures 2 C and D.

Firstly, as we specify from line 143 on, we followed a hierarchical approach to test our hypotheses. Therefore, using a parametric general linear model we projected E/I balances from all five regions onto main behavioural parameters of interest to assess significance in the context of all other regions of interest. The bivariate correlations displayed in the figure were performed if, and only if the GLM yielded a significant main effect of one brain region. The scatterplots displaying the residuals, and the bivariate correlations we report, therefore can (a) be thought of as a post-hoc test and (b) are primarily used for visualization. The GLM is, however, our main basis of inference. The residuals are obtained from a parametric model and measured on a continuous scale. Since they are directly taken from a GLM, ranking data and assessing a monotonic relationship between variables instead of a linear one would mean losing statistical power and important information incorporated in our original model. For our analysis, the distance between measured data and not just their relative rank is of direct relevance.

Secondly, we completely agree with the reviewer that parametric Pearson correlations are more susceptible to outliers. However, there are three things we want to point out: First, again, the main test of interest is the GLM that incorporates all the five E/I balances. The significant results of this model could be confirmed (marginally for 2D) with a robust regression analyses, which accounts for potential outliers (2C: $t_{23} = 2.488$, $p = 0.021$; 2D: $t_{23} = 1.925$, $p = 0.067$). Secondly, to test whether the data points in question in figure 2C/D are indeed outliers, we checked whether they were more than three standard deviations from the group mean. This was not the case for any of the data points. Third, we also re-analyzed the bivariate correlation on the residuals using a robust regression. This yielded the same pattern of results, although the E/I correlation with the cost effect on RT (2D) is now only marginally significant (2C: $t_{27} = 2.665$, $p = 0.013$, 2D: $t_{27} = 2.034$, $p = 0.052$).

To briefly summarize this: (1) Results of our main GLM could be confirmed with a robust regression analysis that already does account for potential outlier effects, (2) the residuals displayed in 2C/D are based on this linear regression and reflect a post-hoc test, (3) the parametric range rather than just ordinal distance is of relevance here, and (4) even when applying robust regression to the

residuals from the GLM, we can recapitulate the same pattern of results.

To make this clear to the reader, we have incorporated the following paragraph into the manuscript in the section “Cortical E/I balance and patch-leaving behaviour”:

” Please note that the results displayed in figures 2C and D primarily serve to illustrate the effects obtained in our main GLM. Nevertheless, inspection of these panels reveals three data points that appear further away from the remainder of the data. Therefore, we have additionally analysed the same data (the residuals of dACC E/I balance and behaviour) with a robust regression analysis. This confirmed the pattern of results reported above (2C: $t_{27} = 2.665$, $p = 0.013$, $CI_{95} = [0.120 - 0.924]$; 2D: $t_{27} = 2.034$, $p = 0.052$, $CI_{95} = [-0.004 - 0.810]$).”

Additionally, we have added the following to our methods section (Regression analyses, page 23):

“For the patch leaving phase (Figure 2C and D) we have additionally analysed the residuals of dACC E/I balance and behaviour with a robust regression analyses with a bisquare weight function (tuning constant = 4.685). “

And we have amended the legends to figure 2C/D and 3B/C/D accordingly:

“Pearson correlation on residuals (compare main text and methods)“for 2C and “ Pearson correlation on residuals” for all remaining figures as well as relevant test statistics.

Point 2: relationship between reported t-values and correlation.

I think the discrepancy between the t-values and the r-values is that the t-values refer to the results of the regression, while the r-values refer to the subsequent partial correlations. It would be good for the authors to clarify, but I think this is not a point for concern.

The reviewer is correct. We have added the following sentence to our methods section to make this relationship clear for readers (page 29, Behavioural parameters and E/I balance):

” We report t- and p-values for each significant regression coefficient ($p < 0.05$), testing for differences from zero, as well as r- and p- values for the subsequent partial correlations (if applicable), both with their respective 95% confidence intervals.”

#Reviewer 2

All of my concerns regarding the regression analyses results are resolved.

We thank reviewer 2 for their comments on previous versions of the manuscript and we are pleased that all concerns could be solved.

Reviewer #3

I have carefully re-read the comments from reviewer #2 and the author's responses in both round of review. I find that the author's went out of their way to explain their statistical approach and gave very clear description of the analyses. The additional analyses using robust regression and Cook's distance to identify outliers are appropriate and underline the author's claims of an effect of dACC E/I balance. Because I also don't know what reviewer #2 communicated to the editors about his remaining concerns, I cannot evaluate, whether these have been addressed by the authors. But from the material that I have at my disposal, I would think that they responded to the comments to the best of their knowledge and abilities and would support a publication of the paper in NCOMMS.

Thank you again for your detailed comments on this and all previous versions of our manuscript.